



# Stress rotation — The impact and interaction of rock stiffness and faults

Karsten Reiter

TU Darmstadt, 64287 Darmstadt, Schnittspahnstraße 9

**Correspondence:** Karsten Reiter (reiter@geo.tu-darmstadt.de)

**Abstract.** It has been assumed, that the maximum compressive horizontal stress ($S_{\mathrm{Hmax}}$) orientation in the upper crust is governed on a regional scale by the same forces that drive plate motion. However, several regions are identified, where stress orientation deviates from the expected orientation due to plate boundary forces (first order stress sources), or the plate wide pattern. In some of this regions a gradual rotation of the $S_{\mathrm{Hmax}}$ orientation has been observed.

Several second and third order stress sources have been identified, which may explain stress rotation in the upper crust. For example lateral heterogeneities in the crust, such as density, petrophysical or petrothermal properties and discontinuities, like faults are identified as potential candidates to cause lateral stress rotations. To investigate several of the candidates, generic geomechanical numerical models are utilized. These models consist of up to five different units, oriented by an angle of 60° to the direction of contraction. These units have variable elastic material properties, such as Young's modulus, Poisson ratio and density. Furthermore, the units can be separated by contact surfaces that allow them so slide along these faults, depending on a selected coefficient of friction.

The model results indicate, that a density contrast or the variation of the Poisson's ratio alone sparsely rotates the horizontal stress orientation ($\leqq 17°$). Conversely, a contrast of the Young's modulus allows significant stress rotations in the order of up to 78°; not only areas in the vicinity of the material transition are affected by that stress rotation. Stress rotation clearly decreases for the same stiffness contrast, when the units are separated by low friction discontinuities (19°). Low friction discontinuities in homogeneous models do not change the stress pattern at all, away from the fault; the stress pattern is nearly identical to a model without any active faults. This indicates that material contrasts are capable of producing significant stress rotation for larger areas in the crust. Active faults that separates such material contrasts have the opposite effect, they rather compensate stress rotations.

## 1 Introduction

The knowledge growth of the in-situ stress tensor in the Earth's crust is an important topic for a better understanding of the endogenic dynamic, seismic hazards or the exploitation of the underground. Therefore, several methods have been developed



to estimate both, the stress orientation or the stress magnitude. Stress orientation data are compiled globally in the World Stress

Map database (Heidbach et al., 2010, 2018; Sperner et al., 2003; Zoback et al., 1989; Zoback, 1992). Based on such data compilations, it was assumed, that patterns of stress orientation on a regional scale are more or less consistent within tectonic plates (Coblentz and Richardson, 1995; Klein and Barr, 1986; Müller et al., 1992; Richardson et al., 1979).

The plate-wide pattern is overprinted on a regional scale by the current collisional systems. Recent examples in Europe are the Alps (Reinecker et al., 2010), the Apennines (Pierdominici and Heidbach, 2012) or the Carpathian Mountains (Bada et al.,

1998; Müller et al., 2010). Closely related to that are inhomogeneities of crustal thickness, density and topography (Artyushkov, 1973; Ghosh et al., 2009; Humphreys and Coblentz, 2007; Naliboff et al., 2012). It was suggested, that remnant stresses due to old plate tectonic events are able to overprint stress orientation on a regional scale (e.g. Eisbacher and Bielenstein, 1971; Richardson et al., 1979; Tullis, 1977). Such old basement structures also provide geomechanical inhomogeneities and discontinuities, which have the potential to disturb the stress field. However, pre-Cenozoic orogens (or 'old' suture zones),

often covered and hidden by (thick) sediments, and were rarely indicated to cause significant stress rotation. In many cases it is the opposite: old orogens have apparent no impact on the present-day crustal stress pattern, e.g. the Appalachian Mountains (Evans et al., 1989; Plumb and Cox, 1987) or Fennoscandia (Gregersen, 1992). Deviations from the assumed stress pattern (stress rotations) are observed recently in several regions, such as in Australia, Germany or Northern America (Heidbach et al., 2018; Reiter et al., 2015; Lund Snee and Zoback, 2018, 2020). However, these effects can only partly explained by the

topography or lithospheric structures.

Complex stress pattern in central-western Europe was a subject of several numerical investigations in the last decades (Grünthal and Stromeyer, 1986, 1992, 1994; Gölke and Coblentz, 1996; Goes et al., 2000; Marotta et al., 2002; Kaiser et al., 2005; Jarosiński et al., 2006). These 2-D models was able to reproduce some of the observed stress pattern, applying variable elastic material properties or discontinuities. However, the drawback of 2-D models are the limitation, that they have to integrate

topography, crustal thickness and stiffness to one property; furthermore, they overestimate horizontal stresses (Ghosh et al., 2006). None of such studies compared the impact of the influencing factors separately.

In order to address the question, which properties are able to cause substantial stress rotations away from the material transition or a discontinuity, simple generic models will be used. They are inspired by the crustal structure and the stress pattern in the German Central Uplands, where the $S_{\mathrm{Hmax}}$ orientations is about 150°. This is in contrast to a north-south orientation (0°)

of $S_{\mathrm{Hmax}}$ to the north and to the south of the uplands (Fig. 1, Reiter et al., 2015). The basement structures there are striking about 30°, which is perpendicular to the observed $S_{\mathrm{Hmax}}$ orientation. The influence of the structures will be tested with variation of the Young's modulus, the Poisson's ratio, the density and low friction discontinuities, which separates the crustal blocks. Each property is tested separately first to avoid interdependencies; possible interactions are tested afterward.





## 2 Stress in the Earth crust

### 2.1 Sources of crustal stresses

The major contribution of stresses in crustal Earth is gravity acceleration. The acting body force is a function of density, the effective gravity and depth. The second major driver are the forces propelling plate tectonics. Plate boundary forces where identified and derive deviatoric stresses (e.g. Bott and Kusznir, 1984; Chapple and Tullis, 1977; Forsyth and Uyeda, 1975; Richardson, 1992; Zoback et al., 1989). The visible products of that are collisional zones or orogens, having a significant topography and a crustal root, which are able to overprint crustal stress pattern on a regional scale, like the fan shape stress pattern in the western and northern foreland of the Alps (Kastrup et al., 2004; Reinecker et al., 2010). There are several other features in the continental crust, which are also able to bias stress pattern on a local or regional scale. The most of these features are a product of previous geodynamical processes, such as passive continental margins, sedimentary basins, density/gravity anomalies, topography, crustal roots, etc.

A systematic classification was developed to range the manifold stress sources, depending on their spatial coverage in first, second and third order stress sources (Heidbach et al., 2007, 2010, 2018; Zoback et al., 1989; Zoback, 1992). First order stress sources extend over a distance of >500 km, which is larger than the thickness of the lithosphere, second order stress sources extend over a distance of $100 - 500$ km, which is approximate the same thickness like the lithosphere, and third order stress sources extend over an distance of <100 km, which is smaller than the thickness of the lithosphere. Second and third order stress sources are able to disturb overall stress orientation trend from regional through local to reservoir scale (Heidbach et al., 2007, 2010, 2018; Müller et al., 1997; Tingay et al., 2005; Zoback, 1992; Zoback and Mooney, 2003).

First order stress sources next to gravity are plate boundary forces: ridge push, slab pull (Bott and Kusznir, 1984; Forsyth and Uyeda, 1975; Richardson and Reding, 1991; Richardson, 1992; Zoback and Zoback, 1981; Zoback et al., 1989; Zoback, 1992), trench suction (Elsasser, 1971), gravitational potential energy (GPE) which is related to the inhomogeneous topography and mass distribution in the lithosphere (Ghosh et al., 2009; Humphreys and Coblentz, 2007; Naliboff et al., 2012) and basal drag or tractions originating from mantle convection (Adams and Bell, 1991; Becker and Faccenna, 2011; Ghosh et al., 2013; Gough, 1984; Lithgow-Bertelloni, 2004; McGarr, 1982; Steinberger et al., 2001).

Second order stress sources are lithospheric flexure due to isostatic compensation or sediment loading on continental margins (Bott and Dean, 1972), membrane stress (Bott and Kusznir, 1984; Turcotte, 1974b) seamount loading, upwarping ocean-ward of the trench (Bott and Kusznir, 1984; Turcotte and Oxburgh, 1973; Walcott, 1970; Zoback, 1992), localized lateral density contrasts/buoyancy forces (Artyushkov, 1973; Bott and Dean, 1972; Fleitout and Froidevaux, 1982), lateral strength contrasts or anisotropy of material properties, topography (Bott and Kusznir, 1984), continental rifting, large fault zones, lateral contrasts of heat production (hydrothermal, volcanic) (Bott and Kusznir, 1979), tensile stress due to cooling of (oceanic) lithosphere (Bott and Kusznir, 1984; Turcotte and Oxburgh, 1973; Turcotte, 1974a), flexural stresses due to deglaciation (Carlsson and Olsson, 1982; Stein et al., 1989; Wu and Johnston, 2000) and mass re-distribution on the surface (sedimentation and erosion processes: Bott and Kusznir, 1984; Haxby and Turcotte, 1976; Turcotte and Oxburgh, 1976).





Third order stress sources are local density- or strength contrasts, internal basin geometry, basal detachments, active faults, remnant stresses as a result of ancient tectonic forces (Eisbacher and Bielenstein, 1971; Richardson et al., 1979; Tullis, 1977) and incised valleys (Savage and Swolfs, 1986). Furthermore, there are man-made increase or decrease of load (e.g. excavations 90 or embankment dams) and downhole pressure changes (production or grouting of fluids).

## 2.2 The stress tensor

Mechanical stress describes the internal forces in solids that neighbouring particles of a continuous material apply on each other. The stress tensor is a second rank tensor, that consists of nine components from which six are independent on account of the symmetry characteristics. Three of them are normal stresses, orthogonal to each other ($\sigma_{xx}$, $\sigma_{yy}$, $\sigma_{zz}$) and three of them are 95 shear stresses ($\tau_{xy}$, $\tau_{xz}$, $\tau_{yz}$). Choosing an optimal reference system, all shear stresses will disappear ($\tau_{xy} = \tau_{xz} = \tau_{yz} = 0$) and the three normal stresses becomes principal stresses. Such principal stresses are independent from the reference system and are denoted in the order of magnitude: $\sigma_1 > \sigma_2 > \sigma_3$. For areas without topography, having lateral homogeneous material properties and density, such as sedimentary basins, it is assumed, that the vertical stress ($S_V$) is a principal stress (Anderson, 1951; Brudy et al., 1997; Herget, 1973; McGarr and Gay, 1978). $S_V$ is a cumulative product of the particular rock density, 100 depth and gravity (Herget, 1973). Consequently, both remaining principal stresses are aligned horizontally to the Earth surface, which are the minimum- and the maximum horizontal stress ($S_{hmin}$ and $S_{Hmax}$ respectively) and again orthogonal to each other. Using that simplification, the stress tensor can be described with the magnitude of $S_V$, $S_{Hmax}$ and $S_{hmin}$ and the orientation of $S_{Hmax}$. The stress regime (Anderson, 1905, 1951) are defined by the relative stress magnitudes, which are normal faulting regime ($S_V > S_{Hmax} > S_{hmin}$; $S_V = \sigma_1$), strike slip regime ($S_{Hmax} > S_V > S_{hmin}$ $S_V = \sigma_2$) and thrust faulting regime ($S_{Hmax} >$ 105 $S_{hmin} > S_V$ ; $S_V = \sigma_3$).

## 2.3 Indicators of stress orientation

Data indicating $S_{Hmax}$ orientation in the Earth' crust are compiled since the 1970's (Hast, 1973; Ranalli and Chandler, 1975; Richardson et al., 1976; Sbar and Sykes, 1973), using fault plane solutions, overcoring, hydraulic fracturing or geological indicators. After recognition that borehole breakouts can be used as an indicator of $S_{Hmax}$ orientation (Bell and Gough, 1979; 110 Bell, 1996a; Hottman et al., 1979; Plumb and Hickman, 1985; Zoback et al., 1985), much more data became available. This led under the International Lithosphere Program to the formation of the World Stress Map (WSM) database. In the first publication, the WSM database comprised 3.574 entries (Zoback et al., 1989) and increased recently to 42.870 entries (Heidbach et al., 2018). The WSM database provide an assignment of qualities of the $S_{Hmax}$ orientation data. The quality criteria range from A to E, where A-quality are the most reliable data, and E-quality contain ambiguous or poorly usable information.

There are three large groups of stress indicators, which are derived from different depth ranges. They are geophysical data (<40 km), borehole data (≪10 km) and geological data from the Earth surface. The database comprises present-day $S_{Hmax}$ orientation data, derived from single fault plane solutions (FMS), average fault plane solutions (FMA) stress inversion based on fault plane solutions (FMF), borehole breakout data (BO, BOT, BOC), hydraulic fracturing (HF, HFG, HFM, HFP), drilling induced tensile fractures (DIF), overcoring (OC) and geological indicators (volcanic alignment – GVA and fault slip analysis –



GFI) and other rare methods. There are many text books and publications available to delve deeply into the methods (Amadei and Stephansson, 1997; Bell, 1996a; Célérier, 2010; Richardson et al., 1979; Schmitt et al., 2012; Zang and Stephansson, 2010; Zoback, 1992).

## 2.4  Stress rotation in the upper crust

The term stress rotation is used to describe the $S_{\text{Hmax}}$ re-orientation vertically (down-well) or rather horizontally i.e. in the map

view perspective. The latter is clearly the focus of this study. All the stress sources interact with each other and therefore stress at a certain point comprises the sum of all stress sources from plate wide to very local stress sources. In the case that a regional stress pattern is disturbed by a local stress source, the difference between the resulting stress orientation and the regional stress source can be described by the angle $\gamma$ (Sonder, 1990). The resulting (counter-)clockwise rotation ($\gamma$) can be substantial and can last in a change of the stress regime (Jaeger et al., 2007; Sonder, 1990; Zoback, 1992).

Shallow stress orientations are mostly consistent with those data, inferred from deep focal mechanism, and therefore a vertical uniform stress field in the brittle crust is assumed (Heidbach et al., 2018; Zoback et al., 1989; Zoback, 1992). However, systematic stress rotations are observed within deep wells (Schoenball and Davatzes, 2017; Zakharova and Goldberg, 2014). This is in particular observed and expected as a result of decoupling by evaporites (e.g. Cornet and Röckel, 2012; Röckel and Lempp, 2003; Roth and Fleckenstein, 2001), or man-made activities in the underground (e.g. Martínez-Garzón et al., 2013;

Müller et al., 2018; Ziegler et al., 2017). However, both is not a subject of that study.

## Density contrast and topography

Variability of density within the crust or lithosphere have a significant impact on the stress state (Artyushkov, 1973; Fleitout and Froidevaux, 1982; Frank, 1972; Ghosh et al., 2009; Humphreys and Coblentz, 2007; Naliboff et al., 2012). Assameur and Mareschal (1995) showed, that local stresses due to topography and crustal inhomogeneities are in the order of tens of MPa,

which are on the similar magnitude as the plate boundary forces. Gravitational forces are also derived by surface topography (Miller and Dunne, 1996; Zoback, 1992). On top of mountains, $S_{\text{Hmax}}$ is oriented parallel to the ridge and perpendicular at the foot of the mountain chain. Along passive continental margins, similar effects like for topography are to observe (Bell, 1996b; Bott and Dean, 1972; King et al., 2012; Stein et al., 1989; Tingay et al., 2005; Yassir and Zerwer, 1997). Sonder (1990) investigated the interaction of different regional deviatoric stress regimes ($\sigma_D$) with stresses arising from buoyancy forces ($\sigma_G$)

and observe a rotation of $S_{\text{Hmax}}$ of up to 90°. According to that, $S_{\text{Hmax}}$ rotates toward the normal trend of the density anomaly. If regional stresses are large, compared to stresses driven by a density anomaly ($\sigma_D/\sigma_G \ll 1$), the influence of density anomaly is small and vice versa: If the regional stress is small, compared to the stress driven by the density anomaly ($\sigma_D/\sigma_G \gg 1$), the impact of a density anomaly on the resulting stress field is large. In the case that both stress sources are on a similar level ($\sigma_D/\sigma_G \approx 1$), small changes of one of the stress sources are able to change the stress regime, and therefore potentially the

deformation style.





**Strength contrast**

Mechanical strength describes the material behaviour under the influence of stress and strain. The focus here is on elastic material properties, characterized by the Young's modulus and the Poisson's ratio. Stress refraction between two elastic media can be calculated, but only at the interface of the two media, based on the known stress state on one side of the interface and
the Young's modulus of both (Spann et al., 1994). Stress rotation due to strength contrast are e.g. reported for the Peace River Arch in Alberta (Adams and Bell, 1991; Bell and Lloyd, 1989; Fordjor et al., 1983). Potential stress rotation is confirmed by several numerical studies (Bell and Lloyd, 1989; Grünthal and Stromeyer, 1992; Mantovani et al., 2000; Marotta et al., 2002; Spann et al., 1994; Tommasi et al., 1995; Zhang et al., 1994).

**Discontinuities**

Discontinuities are planar structures within or between rock units, where the shear strength is (significant) lower than that of the surrounding rock. Genetically, discontinuities can be classified into bedding, schistosity, joints and fault planes. In the context of that study the term discontinuity refers to fault planes or fault zones. Similar to the Earth surface, (nearly) frictionless faults act like a free surface in terms of continuum mechanics (Bell et al., 1992; Bell, 1996b; Jaeger et al., 2007). One of the three principal stresses must be oriented perpendicular to the frictionless fault, the two remaining ones are parallel to the
discontinuity. For this reason, the stress tensor bends near a frictionless fault, depending on its orientation. Significant stress rotation in the context of faults are reported (Adams and Bell, 1991; Bell and McCallum, 1990; Mazzotti and Townend, 2010; Yale, 2003). However, Yale (2003) assumes, that stress rotation occurs several kilometres away from the fault. Small differences between the horizontal stresses increases the effect of faults on the local stress pattern, whereas large stress differences lead to more homogeneous stress pattern (Laubach et al., 1992; Yale, 2003). The impact of faults on stress rotation is investigated by
numerical models (e.g. Homberg et al., 1997; Zhang et al., 1994; Tommasi et al., 1995).

## 3 Regional setting

### 3.1 Stress Orientation in Central Europe

Crustal stress data from Europe have been collected since the 1960's (e.g. Froidevaux et al., 1980; Greiner, 1975; Greiner and Illies, 1977; Hast, 1969, 1973, 1974; Kohlbeck et al., 1980; Ranalli and Chandler, 1975) and later within the framework of
the World Stress Map database. These data were applied to investigate the crustal stress pattern for Western Europe (Ahorner, 1975; Carafa and Barba, 2013; Gölke and Coblentz, 1996; Klein and Barr, 1986; Müller et al., 1992), Scandinavia (Gregersen, 1992), Central Europe (Grünthal and Stromeyer, 1986, 1992, 1994), the Alps (Kastrup et al., 2004; Reinecker et al., 2010), or for the Mediterranean area (Rebaï et al., 1992). As principal $S_{Hmax}$ orientation are identified as north-west to north-north-west in Western Europe, a slightly rotation to north-east in central Europe, a west-north-west orientation in Scandinavia and a
east-west orientation in the Aegean Sea and western Anatolia.





$S_{\text{Hmax}}$ orientation in western Europe of $145°\pm 26°$ deviates clockwise by about $17°$ (Müller et al., 1992) to the direction of absolute plate motion from Minster and Jordan (1978). This is in agreement with Zoback et al. (1989) which obtained a better fit for relative plate motion between Africa and Europa, than for absolute plate motion. As major reasons for the observed stress pattern in western and central Europe are the ridge push of the Mid-Atlantic ridge and the collisional forces along the

southern plate margins are identified (Goes et al., 2000; Gölke and Coblentz, 1996; Grünthal and Stromeyer, 1986, 1992; Klein and Barr, 1986; Müller et al., 1992; Richardson et al., 1979; Zoback et al., 1989; Zoback, 1992).

A fan like stress pattern has been observed in the western Alps and Jura mountains, where $S_{\text{Hmax}}$ in front of the mountain chain is perpendicular to the strike of the orogen. Müller et al. (1992) assumes, that these structures only locally overprints the general stress pattern. However, in the light of the recently available data, it is assumed, that the $S_{\text{Hmax}}$ orientation is rather

controlled by gravitational potential energy of the alpine topography than by plate boundary forces (Grünthal and Stromeyer, 1992; Reinecker et al., 2010). The explanation of that crustal structure are a cold, dense and slowly subsiding lithospheric root beneath the Alps (Müller and Zürich, 1984).

Stress pattern in western and central Europe has been an subject of several modelling attempts in the last three decades (Grünthal and Stromeyer, 1986, 1992, 1994; Gölke and Coblentz, 1996; Goes et al., 2000; Marotta et al., 2002; Kaiser et al.,

2005; Jarosiński et al., 2006). Among other things, it was examined which factors contributes to the observed stress pattern. Investigated was the impact of a stiffness contrast in the crust (Grünthal and Stromeyer, 1986, 1992, 1994; Jarosiński et al., 2006; Kaiser et al., 2005; Marotta et al., 2002), the elastic thickness of the crust (Jarosiński et al., 2006), the stiffness contrast of the mantle (Goes et al., 2000), a lateral density contrast or topographic effects (Gölke and Coblentz, 1996; Jarosiński et al., 2006), the post-glacial rebound in Scandinavia (Kaiser et al., 2005) and activity on faults (Kaiser et al., 2005; Jarosiński et al.,

200  2006).

Stiffness variation in the lithosphere, e.g the Teisseyre-Tornquist Zone (TTZ) or the Bohemian Massif (BM), has been identified as the major reasons for observed stress rotation in Central Europe (Goes et al., 2000; Gölke and Coblentz, 1996; Grünthal and Stromeyer, 1986, 1992, 1994; Kaiser et al., 2005; Marotta et al., 2002; Reinecker and Lenhardt, 1999). One example is the fan shaped stress pattern in the North German Basin (NGB), with a rotation of $S_{\text{Hmax}}$ from north-west in

the western part to north-east in the eastern part of the basin as a product of the TTZ, which is the boundary between the Phanerozoic Europe (Avalonia) and the much stiffer Precambrian Eastern European Craton (Baltica). However, Jarosiński et al. (2006) came to the conclusion, that active tectonic zones and topography has major effects, whereas the stiffness contrast lead only to minor effects. Post-glacial rebound does have an effect on the stress pattern in the NGB (Kaiser et al., 2005). Lateral variation of density do not have a significant impact on the stress pattern (Gölke and Coblentz, 1996), it provides only

local effects. Finally, small differential stress allows significant stress rotation (Grünthal and Stromeyer, 1992).

## 3.2 Basement structures in Germany

The Variscan orogen is a product of the late-Paleozoic collision of the plates Gondwana and Avalonia (Laurussia) in late Devonian to early Carboniferous time, which lead to closure of the Rheic Ocean (Matte, 1986), and finally the formation of the super-continent Pangea. Despite the fact, that the European Varicides are well investigated in the last century and decades (e.g.





Franke, 2000, 2006; Kroner et al., 2007; Kroner and Romer, 2013), it is for example still a matter of debate, whether several microplates have been amalgamated in-between or not.

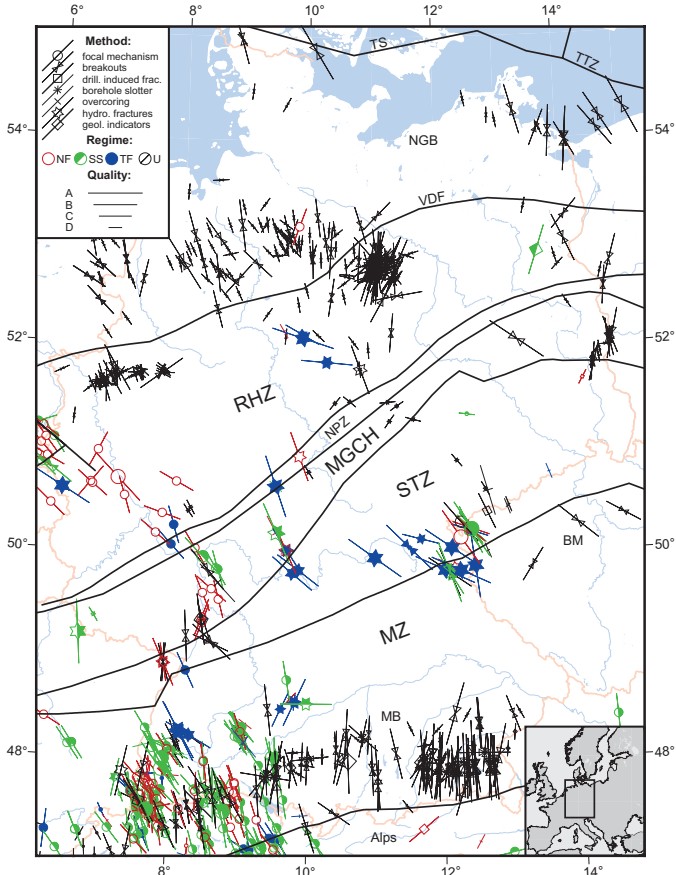

**Figure 1.** Stress orientation in the German Central Uplands with the basement structural elements, political boundaries (red) and the major river (blue). Bars represent orientations of maximum horizontal compressional stress ($S_{\mathrm{Hmax}}$), line length is proportional to quality. Colours indicate stress regimes, with red for normal faulting (NF), green for strike–slip faulting (SS), blue for thrust faulting (TF), and black for unknown regime (U). The Variscan basement structures introduced by Kossmat (1927) are visualized; the regional segmentation is: MZ = Moldanubian Zone, BM = Bohemian Massif, MGCH = Mid-German Crystalline High, NPZ = Northern Pyllite Zone, RHZ = Rheno-Hercynian Zone, STZ = Saxo-Thuringian Zone, and VDF = Variscan Deformation Front. Other structures are: TS = Thor Suture, TTZ = Teisseyre-Tornquist Zone, NGB = North German Basin, MB = Molasses Basin; (Redrawn after Franke, 2014; Grad et al., 2016).

Kossmat (1927) published the structural zonation of the European Variscides, which is still widely used (Fig. 1). The parts north-west of the Rheic Suture Zone are the Rheno-Hercynian Zone (RHZ) with the sub-unit of the Northern Phyllite Zone (NPZ), both with Laurussian origin. South-east of the suture zone are the Mid-German Crystalline High (MGCH), the Saxo-
Thuringian Zone (STZ) and the Moldanubian Zone (MZ); all where exclusively part of Gondwana, except the MGCH.





The Rheno-Hercynian Zone (RHZ) is exposed in the Rhenish Massif, in the Harz mountains and in the Felchting horst. Dominant are Devonian to lower Carboniferous clastic shelf sediments (Franke, 2000; Franke and Dulce, 2017). These low metamorphic slates, sandstones, greywacke and quartzite are supplemented with continental and oceanic volcanic rocks, reef limestones and a few older gneisses. Further to the north of the RHZ are the sub-variscan foreland deposits, consisting of clastic
sediments and coal seams.

The Northern Phyllite Zone (NPZ) is uncovered at the southern edge of the low mountain ranges Hunsrück, Taunus and eastern Harz. Petrological it is probably the greenschist facies equivalent (Oncken et al., 1995) of the Rheno-Hercynian shelf sequence (Klügel et al., 1994), consisting of meta-sediments and within-plate metavolcanic rocks (Franke, 2000).

The Mid-German Crystalline High (MGCH) is open in the Palatinate Forest, Odenwald, Spessart, Kyffhäuser, Ruhla Chrys-
talline (Thuringian Forest) and Flechting Horst. It has been interpreted previously as magmatic arc of the Soxo-Thurigian Zone. But Oncken (1997) assumes that the MGCH is composed from both, Saxo-Thuringian and Rheno-Hercynian rocks. Composition and metamorphic grade varies considerably along-strike of the MGCH (Franke, 2000). It consists of late-Paleozoic sediments, meta-sediments, volcanic rocks, granitoides, gabbros, amphibolite and gneisses.

The Saxo-Thuringian Zone (STZ) is exposed in the Thuingian-Vogtlandian Slate Mountains, Fichtel Mountains, Ore Moun-
tains, Saxonian Granulite Massif, Elbe Valley Slate Mountains, and the Lausitz. It consists of Campro-Ordovician mafic and felsic magmatic rocks, late Ordovician to early Carboniferous marine and terrestrial sediments (Franke, 2000; Linnemann, 2004). These rocks underwent metamorphic overprint up to the early Carboniferous with different metamorphism stage up to eclogite- or granulite facies. These units are interspersed by late- or post-orogenic granites.

The Moldanubian Zone (MZ) is exposed in the Bohemian Massif, the Bavarian Forest, the Münchberg Gneiss Massif, the
Black Forest and the Vosges. They consist of mostly high grade metamorphic crystalline rocks (gneisses, granulite, migmatite) and variscan granites (Franke, 2000).

## 4 Model set-up

### 4.1 Model dimension

The chosen model geometry is inspired by the geometrical situation in the German Central Uplands (Fig. 1), but the overall
intention is a generic model. To make it easy to understand, compass directions are used for model description. The model geometry has a north-south extend of 400 km and 300 km in east-west orientation, with a thickness of 30 km (Fig. 2). In the centre of the model, three diagonal units having a width of 50 km are oriented 30° counter-clockwise from east-west. For each of the three unit, separate material properties can be applied. The most northern and southern block has always basic material properties. Furthermore, a model variation is generated, where the units are separated by contact surfaces, allowing free slip
depending on a chosen friction coefficient.

The lateral element resolution is about 3 km consisting primary of hexahedrons and some wedge elements (degenerated hexahedron). Element resolution into depth ranges from 0.44 km near the surface to about 3.4 km at greatest model depth. The total amount is about 166 k elements; the model version having contact surfaces uses 1725 contact elements along each contact





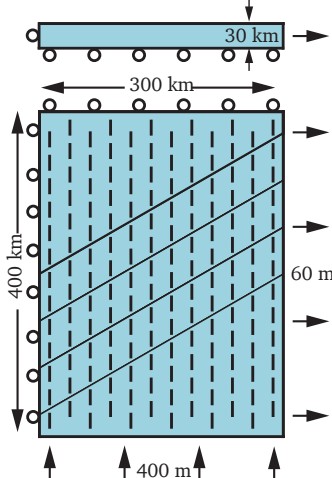

**Figure 2.** Basic model with the applied boundary conditions. The model has a lateral extend of $300 \times 400$ km and a thickness of 30 km. The model consists of five connected units; here visualized in blue, they have all the basic material properties (Tab. 1). The boundary conditions ban motion in x-direction on the western side, in y-direction on the northern side and in z-direction at the model base. A push of 400 m from the south and a pull of 60 m to the east is applied. The resulting $S_{\text{Hmax}}$ orientation (north-south) in a depth of 1000 m are illustrated by the black bars. The four diagonal boundaries can be used as discontinuities.

surface. The mesh is generated using HyperMesh® v.2019. The equilibrium of forces (body forces and boundary condition) is
estimated numerically using the Abaqus®/Standard v.6.14-1 finite element software.

## 4.2   The Finite Element Method

The finite element method (FEM) is used since the 1950's to investigate the stability of structures such as wings of an airplane (Turner et al., 1956). Since the 1970's the method has been introduced in geoscience, to investigate the stress orientation in the Earth crust on certain structures using generic 2-D models (Stephansson and Berner, 1971), or to investigate the stress
orientation pattern for large regions (Richardson et al., 1976; Grünthal and Stromeyer, 1986). The usage of 3-D FEM models, to investigate the stress state in the crust is now a well-established technique (e.g. Buchmann and Connolly, 2007; Hergert and Heidbach, 2011; Hergert et al., 2015; Reiter and Heidbach, 2014). The major reason that complex 2-D or 3-D models can be computed, is the opportunity to use unstructured meshes.

The method in general computes the equilibrium of stresses with forces (boundary conditions), body forces (gravity) and
used material properties. Numerically, this is implemented by partial differential equations:

$$\frac{\delta \sigma_{ij}}{\delta x_j} + \rho x_i = 0 \tag{1}$$

where $\delta \sigma_{ij}$ is the variation of total stress, $\delta x_j$ the geometrical change, and $\rho x_j$ represents the weight of the rock section ($\rho = $ density). The previously described balance of forces is well described by the linear elastic material properties (Hooke's law).





Two material properties, the Young's modulus ($E$) and the Poisson's ratio ($\nu$) are essential; density is not absolutely necessary
but enables that body forces act. The stress state in this study will be calculated based on defined displacement boundary
conditions.

### 4.3  Material properties

The main subject of that study is to investigate the impact of the variation of elastic rock properties, density and friction along
faults on stress orientation in the upper crust. To do this each parameter is tested individually. Figure 3 visualize the range of
density ($\rho$), Young's modules ($E$) and Poisson's ratio ($\nu$) of representative rocks, taken from a textbook (Turcotte et al., 2002).

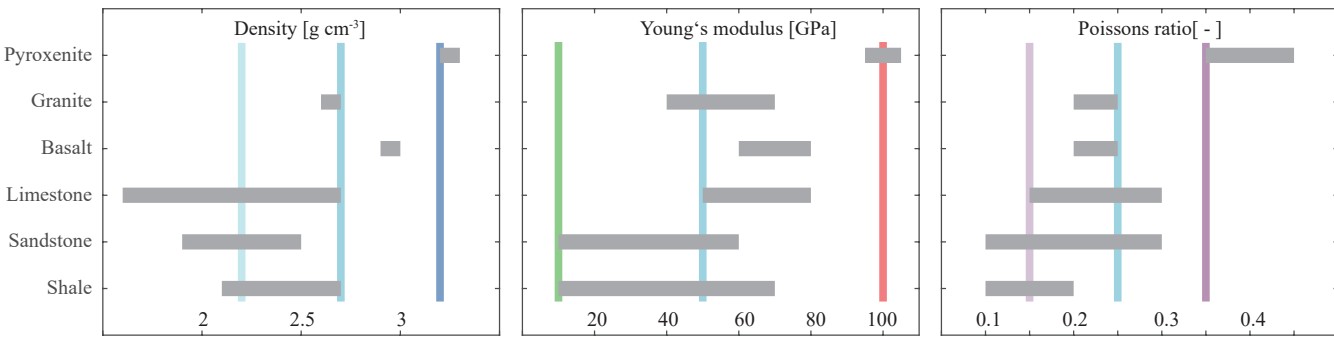

**Figure 3.** Selection of common elastic rock properties (Young's modulus and Poisson's ratio) and density (Turcotte et al., 2002). Coloured
vertical bars indicate applied material properties.

The basic material for this investigation has a density of $\rho = 2.7\,g\,cm^{-3}$, a Poisson's ratio of $\nu = 0.25$ and a Young's modulus
of $E = 50\,GPa$. Such a material could represent for example granite or limestone. Based on this basic material, always a lower
and higher material value is defined (Tab. 1), which is within the range of common rocks properties (Fig. 3). The material with
a low density ($\rho = 2.2\,g\,cm^{-3}$) may represent sediments (sandstone, limestone, shale etc.), where the high density material
($\rho = 3.2\,g\,cm^{-3}$) could represent a rock from the lower crust or the upper mantle. A low Poisson's ratio ($\nu = 0.15$) may
represent sediments (sandstone or shale), and a high Poisson's ratio ($\nu = 0.35$) could represent ultramafic rock. Soft material
with a low Young's modulus ($E = 10\,GPa$) may represent sediments, pre-damaged or weathered rock. Again ultramafic rock
is an example for a stiff rock, having a large Young's modulus ($E = 100\,GPa$).

Laboratory rock experiments in the past delivered friction coefficients of about $\mu = 0.6$ to $0.85$ (Byerlee, 1978). However,
recent investigations using realistic slip rates for earthquakes decreased estimated friction coefficients by one order of mag-
nitude up to $\mu < 0.1$ (Di Toro et al., 2011). Faults are represented by cohesionless contact surfaces in the models. The used
friction coefficients are 0.1, 0.2, 0.4, 0.6, 0.8 and 1.0, which finally covers slow- and fast slip rates as well.





**Table 1.** Material properties and densities which has been used in the models. Bold numbers indicate the properties used, which differ from those of the basic material.

| Name | Young's Modulus $[GPa]$ | Poisson's ratio [-] | Density $[g\,cm^{-3}]$ |
|---|---|---|---|
| Basic material (B) | 50 | 0.25 | 2.7 |
| Low Density (g) | 50 | 0.25 | **2.2** |
| High Density (G) | 50 | 0.25 | **3.2** |
| Low Poisson (p) | 50 | **0.15** | 2.7 |
| High Poisson (P) | 50 | **0.35** | 2.7 |
| Low Stiffness (e) | **10** | 0.25 | 2.7 |
| High Stiffness (E) | **100** | 0.25 | 2.7 |
| Upper Mantle | **130** | 0.25 | **3.25** |

## 4.4 Boundary conditions

The overall $S_{\mathrm{Hmax}}$ orientation on an imagined profile along longitude $11°$ (Fig. 1) displays a north-south orientation in the
North German Basin (NGB) and in the Molasses Basin (MB) north of the Alps, except the Variscan basement units in-between.
According to that, a north-south orientation of $S_{\mathrm{Hmax}}$ is intended for the basic model. To define appropriate boundary conditions
(lateral strain), results from a virtual well in the model centre are compared with data from deep wells. Several strain scenarios
where applied to the pre-stressed basic model. An extension of $60\,\mathrm{m}$ ($\epsilon_x = 0.02$) in east-west direction and a shortening of
$400\,\mathrm{m}$ ($\epsilon_y = -0.1$) in north-south direction (Fig. 2) provides a good fit to stress magnitudes from selected deep wells (Fig. 4,
Brudy et al., 1997; Hickman and Zoback, 2004; Lund and Zoback, 1999). By fitting the data, the focus was more on the
observed $S_{\mathrm{hmin}}$ magnitudes and to a less extend on the $S_{\mathrm{Hmax}}$ magnitudes. The latter are less reliable, as they are usually not
measured; they are calculated on the basis of several assumptions.

## 4.5 Model scenario's

The model geometry consists of five units (Fig. 2). The northern- and southern most block uses always the basic material
properties (Tab. 1). In between are three diagonal units, where material properties or friction properties will be varied. The
numerical lower (L) or higher (H) material properties regarding the basic material (B) will be varied in the following way:
LLL, HHH, LBL, BLB, etc. When the model geometry mimics discontinuities using contact surfaces, all contacts have the
same friction coefficient. The visualization takes place in the following way, where '|' indicates the contact. For example,
HLH with four contacts is |H|L|H|. The $S_{\mathrm{Hmax}}$ orientation will be visualized for a depth of $1000\,\mathrm{m}$ below the surface using a
pre-defined pattern, where the lateral distance to the material transition or discontinuity is $>12.5\,\mathrm{km}$, as far field effects are the
major interest of that study.





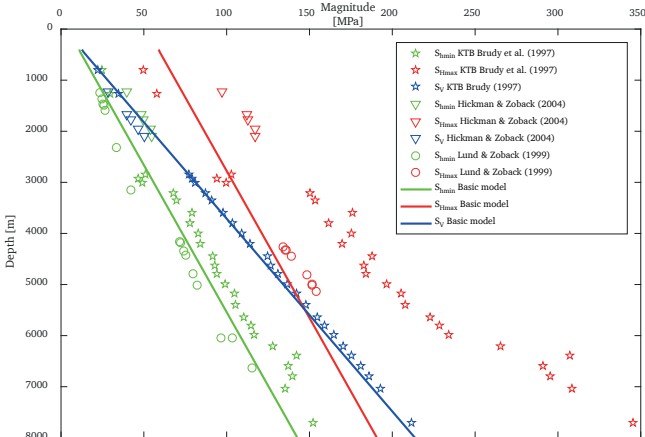

**Figure 4.** Stress magnitudes are potted versus depth. The stress components from the virtual well in the centre of the basic model are shown by the lines, using the boundary conditions illustrated in Fig. 2. Due to the applied initial stress conditions, the stress regime switches from thrust faulting in a depth of around 400 m blow the surface to strike slip faulting, and finally to a normal faulting regime in a depth grater then 5500 m. Additionally, published stress magnitude data are shown for comparison (Brudy et al., 1997; Hickman and Zoback, 2004; Lund and Zoback, 1999).

The variation of the density, the Poisson's ratio, the Young's modulus, the friction coefficient will be tested first. Additionally, the Variation of Young's modulus using low friction contacts, a modification of the model with an additional 30 km stiff mantle and a thinner model, having a thickness of only 10 km will be tested. The latter are only mentioned in the discussion.

## 5 Resulting stress rotation

### 5.1 Density influence

To identify the influence of a density variation, the basic density ($\rho = 2.7\,g\,cm^{-3}$) in blue are varied using a small density (g: $\rho = 2.2\,g\,cm^{-3}$), which is coloured in light blue and a large density (G: $\rho = 3.2\,g\,cm^{-3}$), which is dark blue in Fig. 5.

The low density anomaly (ggg) result in a slightly counter-clockwise ($-6°$) $S_{\mathrm{Hmax}}$ orientation in the basic material near the anomaly (Fig. 5). Within the low density units near the basic material, nearly no rotation is to observe ($-1°$), but turns more counter-clockwise ($-8°$) in the centre of the material anomaly. The angular variation of $S_{\mathrm{Hmax}}$ crossing the units is in the order of $7°$. The high-density anomaly (GGG) results in a slightly clockwise rotation ($+7°$) in the basic material near the anomaly. In the high density unit near the basic material, $S_{\mathrm{Hmax}}$ is minimally influenced ($+1°$), but rotates further clockwise ($+12°$) in the centre of the anomaly. Based on that, the variation across the units is about $11°$. The models with mixed densities in the thee units show a clockwise rotation ($+10°$) of $S_{\mathrm{Hmax}}$ within the lighter material next to the more dense units. The high density units show a counter-clockwise rotation ($-7°$) next to the low density unit; therefore, the total variation of $S_{\mathrm{Hmax}}$ is $17°$.




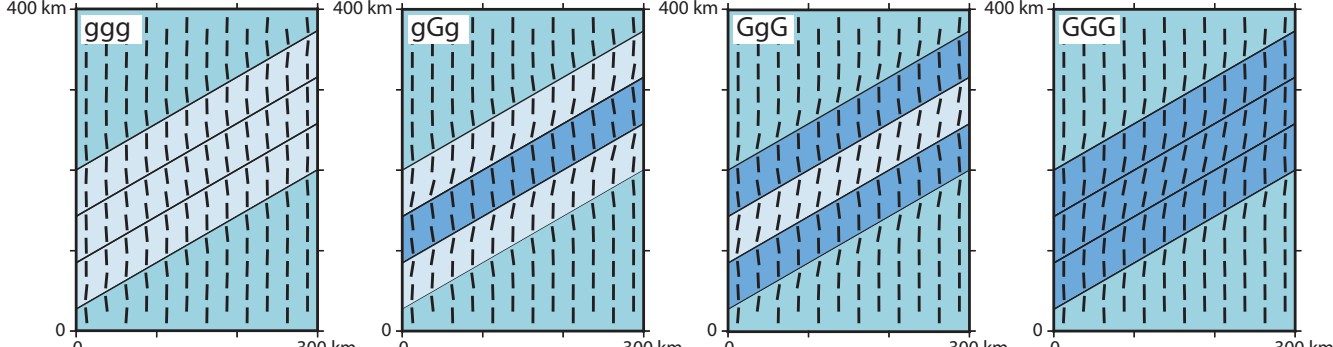

**Figure 5.** Influence of density on the stress orientation. Black bars represents the orientations of the maximum horizontal compressional stress ($S_{\mathrm{Hmax}}$) at a depth of 1000 m. Colours indicate the used material properties. The blue area uses the basic material properties ($\rho = 2.7\,g\,cm^{-3}$), the light blue material uses a lower density (g: $\rho = 2.2\,g\,cm^{-3}$), the dark blue a larger density (G: $\rho = 3.2\,g\,cm^{-3}$).

In general, $S_{\mathrm{Hmax}}$ is oriented parallel to the anomaly in low density units and perpendicular to the anomaly in the large density units. In the centre of the low density units (ggg), the stress orientation becomes perpendicular to the overall structure. For the centre of the high density units (GGG) happens the opposite, $S_{\mathrm{Hmax}}$ becomes parallel to the spacious structure.

### 325    5.2    Influence of the Poisson's ratio

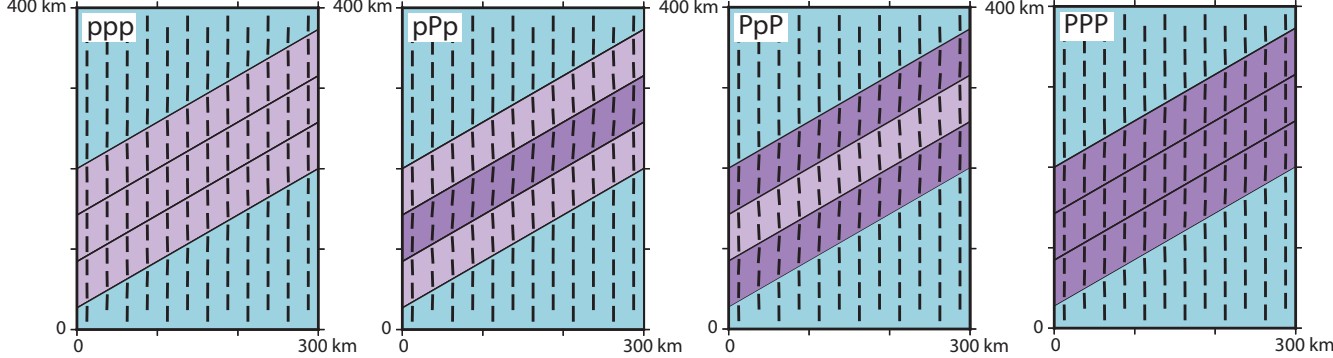

**Figure 6.** Influence of the Poisson's ratio on the stress orientation. Black bars represents the orientations of the maximum horizontal stress ($S_{\mathrm{Hmax}}$) at a depth of 1000 m. Colours indicate the used material properties. The blue area uses the basic material properties ($\nu = 0.25$), the light purple material uses a low Poisson's ratio (p: $\nu = 0.15$), where the dark purple material have a large Poisson's ratio (P: $\nu = 0.35$).

The influence of the Poisson's ratio on the stress rotation is tested by variation of the basic Poisson's ratio ($\nu = 0.25$) using a lower one (p: $\nu = 0.15$) in light purple and a larger one (P: $\nu = 0.35$) in dark purple.

The models with only a lower (ppp: $-1.5°$) and only a higher Poisson's ratio (PPP: $+2.2°$) shows only little $S_{\mathrm{Hmax}}$ rotation (Fig. 6). Mixed models with largest Poisson's ratio variation (pPp and PpP) have some counter-clockwise rotation in the low





Poisson's ratio units ($-3.0°$) and a clockwise rotation in the high Poisson's ratio units ($+4.2°$). Therefore, the total variance of $S_{\mathrm{Hmax}}$ is about $7.5°$.

### 5.3   Impact of Young's modulus

The impact of the Young's modulus variation is investigated taking a basic material (B: $E = 50\,\mathrm{GPa}$) in contrast to a softer material (e: $E = 10\,\mathrm{GPa}$) in green and a stiffer material (E: $E = 100\,\mathrm{GPa}$) in red (Fig. 7).

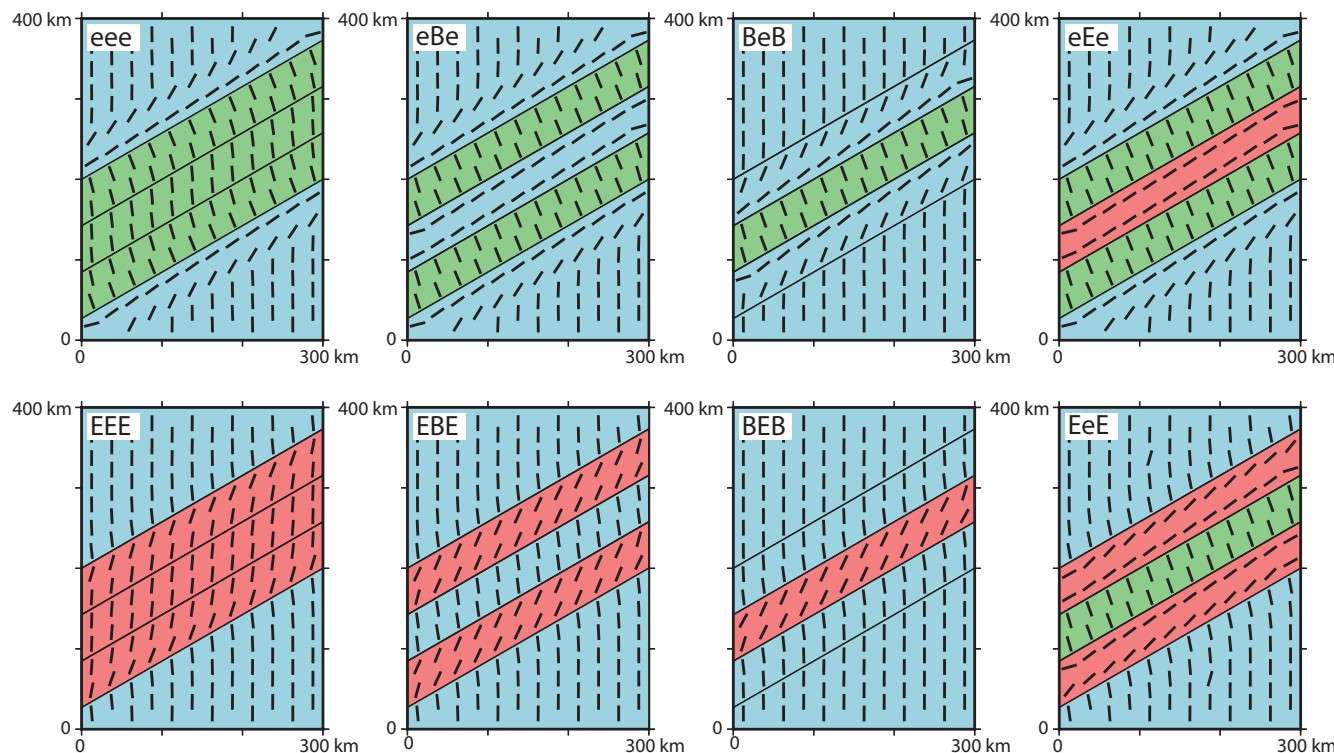

**Figure 7.** Influence of Young's modulus on the stress orientation. Black bars represent the orientations of the maximum horizontal stress ($S_{\mathrm{Hmax}}$) at a depth of 1000 m. Colours indicate the used Young's modulus; the blue area uses the basic material properties (B: $E = 50\,\mathrm{GPa}$), the green material uses a low Young's modulus (e: $E = 10\,\mathrm{GPa}$), where the red material have a large Young's modulus (E: $E = 100\,\mathrm{GPa}$).

The models with the soft units (eee, eBe and BeB) exhibit a strong clockwise $S_{\mathrm{Hmax}}$ rotation ($+56°$) in the units with the basic material and a counter-clockwise rotation in the softer units ($-22°$) near the material transitions (Fig. 7). Within the models having three soft units (eee) the $S_{\mathrm{Hmax}}$ orientation degreases to $-5°$ in the centre of the units. That means within the soft units $S_{\mathrm{Hmax}}$ variation is considerable ($17°$). The total variation is $78°$. The models with the stiff units (EEE, EBE and BEB) exhibits a gentle counter-clockwise rotation in the units with the basic material ($-5.5°$ to $-7°$) next to the stiff units. Within

the stiff units, a significant clockwise rotation ($+20°$ to $+25°$) is apparent next to the basic units. The model having three stiff units (EEE) the $S_{\mathrm{Hmax}}$ orientation degreases to ($+5°$) in the centre. This is a $S_{\mathrm{Hmax}}$ variation of considerable $15°$ within the




stiff units. The total variation is $31°$. For the models with alternating units with soft and stiff material (EeE and eEe), the soft units exhibits a counter-clockwise $S_{Hmax}$ rotation ($-19°$ to $-22°$), where the stiff units displays a clockwise rotation ($+53°$ to $+56°$). Consequently the total variation between the soft and stiff units is $72°$ to $78°$. The general observation is, that next to
the material transition, $S_{Hmax}$ rotates perpendicular to the anomaly for the weak units and parallel for the stiff units.

## 5.4   Influence of faults

Several models with the basic material properties separated by three discontinuities (|B|B|B|) having a friction coefficient ($\mu$) from 0.1 to 1 are tested. The low friction coefficient ($\mu = 0.1$) leads to a counter-clockwise $S_{Hmax}$ rotation of only $-3°$ (Fig. 8). By increasing the friction coefficient to $\mu = 0.2$, the $S_{Hmax}$ rotation is $-2°$, for $\mu = 0.4$, $S_{Hmax}$ rotation is $-1°$. For larger
friction coefficients the $S_{Hmax}$ rotation is below $-1°$. As the $S_{Hmax}$ rotation is too small for a visual differentiation, only the $\mu = 0.1$ model is shown in Figure 8.

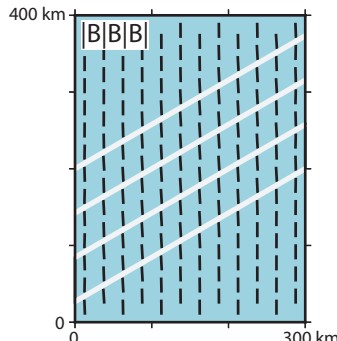

**Figure 8.** Influence of faults on the far field stress orientation. Black bars represents the orientation of the maximum horizontal compressional stress ($S_{Hmax}$) at a depth of $1000\,\mathrm{m}$. All areas have the basic material properties (Tab. 1.). White lines indicate cohessionless discontinuities (faults). The model using a friction coefficient of $\mu = 0.1$ along the three discontinuities is shown. The other models with a larger friction coefficient (up to 1 and larger) have similar results, they are waived out because of the visual similarity.

## 5.5   Stiffness variation combined with low friction faults

The interaction of a significant Young's modulus difference is tested in combination with a low friction coefficient ($\mu = 0.1$) along all four discontinuities. The model with three stiff units (|E|E|E|) provides only little counter-clockwise rotation ($-4°$) in
the basic material near the material transition (Fig. 9). Similar clockwise rotation occur in the stiff units ($+4°$) near the material transition, which decreases to the material centre ($+1°$). However, the total $S_{Hmax}$ variation of about $8°$.

The model with the soft units and the low friction discontinuities (|e|e|e|) displays significant larger rotation then for the stiff units. Clockwise rotation of $+19°$ occurs in the basic material and counter-clockwise rotation of $-13°$ in the soft units. This decreases toward the centre of the soft units ($-9°$). Both rotations together result in about $32°$.





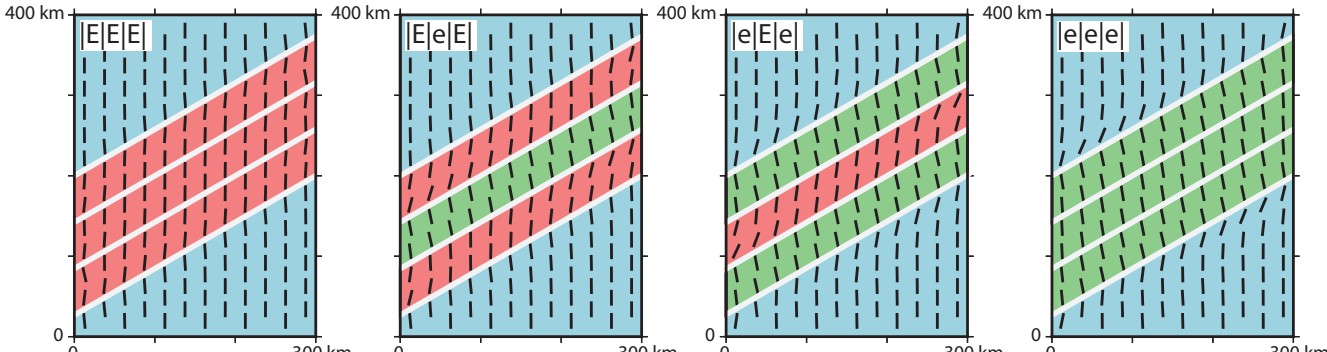

**Figure 9.** Influence of Young's modulus in interaction with low friction faults on the far field stress orientation. Black bars represents the orientations of the maximum horizontal stress ($S_{\text{Hmax}}$) at a depth of $1000\,m$. Colours indicate the used material properties. The blue area uses the basic material properties, the green material uses a low Young's modulus, where the red material has a larger Young's modulus, see Tab. 1. White lines indicate cohessionless discontinuities (faults) with a friction coefficient of $\mu = 0.1$.

In the models with the alternating stiffness with the low friction discontinuities (|E|e|E| and |e|E|e|) provides a counter-clockwise rotation of about $-10°$ to $-12°$ in the soft units. Within the stiff units, the $S_{\text{Hmax}}$ orientation is in the range of $+2°$ to $+7°$. The total variation is up to $19°$.

# 6 Discussion

## 6.1 Model simplification

This study investigates the influence of elastic material properties, density and the friction coefficient on vertical faults on the orientation of $S_{\text{Hmax}}$. Although the model may be inspired by a particular region, but the goal is to gain a better understanding of the interaction of the variable material properties on the stress orientation ($S_{\text{Hmax}}$) . The focus is not on stress rotation close ($<5\,km$) to the material transition or discontinuity, the priority is on the far field effects. Fore sure it is really unlikely that such constant materials with such a thickness exists somewhere in the crust. Not only the geometry is a strong simplification, but 370 also the neglect of various rheological processes in the crust by applying linear-elastic material laws is a strong simplification. However, the overall geometry seems reasonable, as the brittle domain or elastic thickness of the crust ($Te$), which is a measure of the integrated strength of the lithosphere, is in the order of $30\,km$ and more in central Europe (Tesauro et al., 2012). The Moho depth in central Europe is also about $30\,km$ (Grad and Tiira, 2009). Jarosiński et al. (2006) for example used a range of $Te = 30-100\,km$ for their model of central Europe.

The models were tested with an additional stiff mantle with a thickness of $30\,km$. This had no influence on the observed stress pattern in a depth of $1000\,m$ depth. But the models with the same geometry and a total thickness of only $10\,km$ resulted in much lower stress rotation. Therefore, the elastic thickness of the crust and the width of the anomaly is an important constraint




for the possible stress rotation. The depth at which the stress orientation is plotted is also important, as the stress rotation changes with depth.

### 6.2 Stress rotation by density contrast

The lateral variation of the density is responsible for $S_{Hmax}$ rotation in the range of 7° to 17° (Fig. 5). In general, the $S_{Hmax}$ rotates in the low density units slightly toward parallel to the high density unit, whereas $S_{Hmax}$ rotates in the high density units a little bit in the direction to the low density units. Taking a broad range of sediments into account (evaporates, shale, sand- or limestone), they could have even a lower density then the used lowest value ($\rho = 2.2\,g\,cm^{-3}$). However, sediments could reach a thickness of several thousand meters, but not in the order of the model size of 30 km or with such a low density due to increasing compaction with depth. Therefore, the impact of density variation on the stress orientation in nature will be much smaller, or on a very local scale. This agrees with the results of Gölke and Coblentz (1996). Observed significant stress rotation are may be a product of other parameters, observation close to the material transition, or low differential stresses.

### 6.3 Stress rotation due to a variation of the Poisson's ratio

Model results suggest, that the Variation of the Poisson's ratio can be responsible for a $S_{Hmax}$ rotation of about 7.5° (Fig. 6). This is below the uncertainties of stress orientation estimations. Therefore, the variation of the Poisson's ratio can be neglected. It was also no literature to detect, investigating that subject.

### 6.4 Stress rotation due to different Young's modulus

The lateral variation of the Young's modulus can lead to significant $S_{Hmax}$ rotation (Fig. 7). For the used geometry and material parameters, the relative rotation are up to 78°, which is not far from the maximal possible rotation of 90°. The largest rotation occur in the models with the lower Young's modulus, for example the eee model have a total rotation of 78°, whereas the EEE model provides only 31°. This is not surprising as the Young's modulus is simply a measure of the stiffness. Therefore, largest stress rotation due to stiffness contrast will happen in the soft units not in the rigid ones.

$S_{Hmax}$ will be oriented parallel to the structure for stiff units and perpendicular to weak units, which agrees with the literature (Bell, 1996b; Zhang et al., 1994). The largest stress rotation occurs nearest to the material transition and decreases with distance to the material transition, similar to other models (Spann et al., 1994). The importance of stiffness differences is an result of other models too (Grünthal and Stromeyer, 1992; Mantovani et al., 2000; Marotta et al., 2002; Spann et al., 1994; Tommasi et al., 1995). In contrast to that, Jarosiński et al. (2006) found, that a stiffness contrast has only minor effects. But they did not test the stiffness contrast separately; they applied that only in combination with active faults in-between the units. However, this agrees with the results of this study, as faults balance stress rotation by stiffness contrast.

Substantial stress rotations are not observed along major Pre-Mesozoic boundaries and sutures in the eastern United States, like the Greenville front, a suture from Missouri to New York, or in the Appalachian Mountains (Zoback, 1992). Gregersen (1992) reports the same from Fennoscandia. In the case that these tectonic boundaries did not provide a significant stiffness





transition, it is not a contradiction to this study. The observed radial stress pattern southward of the Bohemian massif (Reinecker

and Lenhardt, 1999) agrees well with this study, where $S_{Hmax}$ is perpendicular in the soft sediments of the Upper and Lower

Austrian basin directed to the stiff crystalline Bohemian Massif. More ambiguous would that be for the fan shaped pattern in

western and northern part of the Alpine molasses basin (Grünthal and Stromeyer, 1992; Kastrup et al., 2004; Reinecker et al.,

2010). As reasons, a lateral stiffness contrast of the rock could play a roll, next to the topographic features of the mountain

chain and the overall crustal structure.

Furthermore, important is the depth of observed stress rotation. For example, data in the north-western Alps are dominant

focal mechanisms and in the north-eastern Alps the majority of data are from wells, which are more shallow (Reinecker et al.,

2010).

## 6.5 Comparison of stress rotation due to elastic material properties

The rotation of $S_{Hmax}$ perpendicular (counter-clockwise) to the structure can be observed in material with a lower Young's

modulus next to a material transition most clearly. Rotation in the same direction, but with a less amount is observed in rocks

with a larger density or a smaller Poisson's ratio. Within the units having a larger Young's modulus, $S_{Hmax}$ rotates significant

parallel (clockwise) to the material transition. Similar rotation with a smaller magnitude can be observed in the low density

units or in the units with a larger Poisson's ratio. As rocks with a larger Young's modulus will usually have a larger density and

vice versa (Fig. 3), real rocks will have less $S_{Hmax}$ rotation as suggested by this generic models.

## 425 6.6 Effect of faults on stress orientation

According to the model results, the influence of low friction faults can be neglected concerning the orientation of the far field

stress pattern for homogeneous units. The low friction faults lead to only $3°$ $S_{Hmax}$ rotation in a distance of about 12.5 km next

to the fault zone. This is not in contrast to the strong stress perturbation, observed in the vicinity of faults as one of the three

principal stresses must be oriented perpendicular to a fault, the two remaining ones are parallel to the discontinuity (Bell et al.,

1992; Bell, 1996b; Jaeger et al., 2007). Observations from outcrops investigations indicates stress perturbation within 2 km

(Petit and Mattauer, 1995) or less than 1 km to a fault (Rispoli, 1981); larger stress perturbation is to observe at the termination

of the fault (2-3 km). If $S_{Hmax}$ is parallel next to the fault, it will rotate by $90°$ at the fault termination (Osokina, 1988; Rispoli,

1981).

Yale (2003) suggests significant stress rotation as a product of active faults within a distance of several hundred meters

for large differential stress provinces and several kilometres for regions with small differential stresses. This is supported by

observed stress rotations near a fault within a range of a few hundred meters to a few kilometres (Brudy et al., 1997; Yassir

and Zerwer, 1997). However, not all observed stress rotation agrees to the presented models, like observations offshore eastern

Canada (Adams and Bell, 1991; Bell and McCallum, 1990) where stress rotation occur in a distance of about 10-15 km to a

fault.

Numerical models investigating stress rotations near a fault provides stress rotation between $20°$ and $60°$, next to the fault,

depending on the fault strike, the boundary conditions and the friction or weakness of the fault. Near the termination of the





fault, stress rotation increases to 50-90° (Homberg et al., 1997; Tommasi et al., 1995; Zhang et al., 1994). However, rotation is observed within 2-3 elements away from the discontinuity, which are anyway needed to distribute the deformation in such numerical models. To avoid this, the orientations of $S_{\text{Hmax}}$ in this study is displayed at least four elements away from the contact

surface. In addition, FEM models are unsuitable for representing complex stress-strain patterns near the fault termination if no sufficiently high resolution mesh is available.

### 6.7   Effect of faults combined with stiffness contrasts on stress orientation

The models with low friction faults and a variable stiffness (Fig. 9) illustrate much lower stress rotation then the models without the faults (Fig. 7). It seems to be that discontinuities play an important role to reduce stress rotation produced by lateral Young's

modulus variation. Regarding to the used model geometry and materials, the $S_{\text{Hmax}}$ variation is reduced for the soft models from 78° to 32°, for eee and |e|e|e| in Figs. 7 and 9, using a friction coefficient of $\mu = 0.1$. Also for the mixed models, a change from 78° to 19° is significant. Much lower is the rotation for the stiffer model, with a reduction from 31° to 8° rotation (EEE to |E|E|E|).

    The interaction of a variable Young's modulus and presence of low friction faults is visible in Fig. 10. The observed stress

rotation strongly depends on the depth. In the soft units, $S_{\text{Hmax}}$ rotates counter-clockwise near the surface (0 to -8 km). In contrast to that a clockwise rotation can be observed in greater depth ($18-30$ km).

    The likelihood of seismicity or faults near the interface between stiff and soft units is larger, since differential stresses are greatest there. This fits with the observation of concentrated intraplate earthquakes around cratons (Mooney et al., 2012). On a smaller scale this has been observed for stiff sedimentary layers or rigid dykes, which attracts the occurrence of seismicity

(Roberts and Schweitzer, 1999; Ziegler et al., 2015).

### 6.8   Comparison of model results with observed stress orientations

#### 6.8.1   Variscan basement in Germany

Comparison of modelling results with the stress orientation in the German Variscides is only possible to a certain limit, as the model did not reproduce the detailed structural features of the basement structures and certainly not the (partly) overlying

sediments. The comparison will concern only the results of the models investigating the Young's modulus, as this material parameter have the strongest impact. Taking the structural zonation of the European Variscides of Kossmat (1927) into account, cumulative allocation of geomechanical properties is needed to compare the model results. The Rheno-Hercynian Zone (RHZ) and the Northern Phyllite Zone (NPZ) are dominated by clastic shelf sediments with an low- or mid-metamorphic overprint, which are slate (RHZ) and phyllite (NPZ). This zone, the RHZ and the NPZ together, is the weakest, and will have the

lowest Young's modulus. The Mid-German Crystalline High (MGCH) consists of granitoids or gabbros and their metamorphic equivalents (gneiss, amphibolite), meta-sediments and some volcanites. Therefore, this zone is a stiff unit. The Saxo-Thuringian Zone (STZ) is dominated by meta-sediments, mafic and felsic magmatites and their metamorphosed equivalents, and some high-grade metamorphic rocks (granulite, eklogite). Taking all the different rock types into account, the STZ is more stiff as





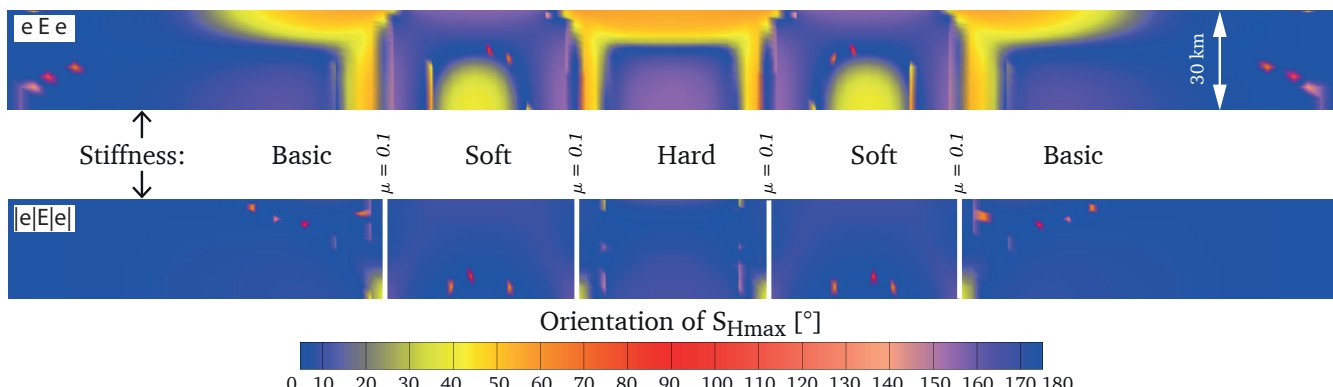

**Figure 10.** North-south depth profiles displaying the $S_{\text{Hmax}}$ orientation colour-coded for a models with variable Young's modulus. In the model without the discontinuities (eEe), $S_{\text{Hmax}}$ is oriented around 40° in the stiffer units next to the softer units near the Earth surface. A similar orientation can be observed in the soft units in the deepest parts. In contrast to that, in the model with the same material properties, but low friction faults (|e|E|e|), the $S_{\text{Hmax}}$ orientation is nearly north-south for all units and depths. (Small coloured dots are artefacts.) The discontinuities with a low friction coefficient counterbalances stress rotations by stiffness contrasts.

the RHZ and weaker than the MGCH. Mechanical, the Moldanubian zone (MZ) can be represented by high-grade metamorphic

rocks (gneiss, granulite, migmatite) and granitoids and will be a stiff unit, similar to the MGCH. Therefore, the units are from the deformable to the rigid ones: RHZ < STZ < MGCH ≈ MZ. According to the model results (Fig. 7), the $S_{\text{Hmax}}$ orientation in the RHZ should deflected counter-clockwise, in the STZ slightly counter-clockwise, and clockwise rotation in the MGCH and MZ.

**Table 2.** Material properties used for the Variscan basement units. The properties are estimated based on Turcotte et al. (2002).

| Vasiscan units | Density $\rho$ $[g\,cm^{-3}]$ | Young's modulus $E$ [MPa] | Poisson's ratio $\nu$ [ ] |
|---|---|---|---|
| Rheno-Hercynian (RHZ) | 2.10 | 20 | 015 |
| N. Phillyite (NPZ) | 2.20 | 30 | 0.15 |
| Mid-German C. (MGCH) | 2.75 | 70 | 0.30 |
| Saxo-Thuringian (STZ) | 2.60 | 50 | 0.25 |
| Moldanubian (MZ) | 2.75 | 70 | 0.30 |

To do this, adoption of the model with dimensionally appropriate elastic material properties would be the best option. These

material properties are estimated based on typical rock values (Tab. 2 Turcotte et al., 2002). The same initial stress procedure




and the same boundary condition and visualization as well is applied. The resulting $S_{\mathrm{Hmax}}$ orientation is visualized in Fig. 11 and will be compared with observed $S_{\mathrm{Hmax}}$ orientation in Fig. 1.

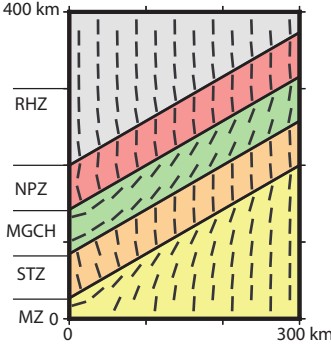

**Figure 11.** Application of estimated material properties of the Variscan units (Tab. 2). Black bars represents the orientations of the maximum horizontal stress ($S_{\mathrm{Hmax}}$) at a depth of 1000 m. The equivalent regions are the RHZ = Rheno-Hercynian Zone, NPZ = Northern Phillyite Zone, the MGCH = Mid-German Crystalline High, the STZ = Saxo-Thuringian Zone and the MZ = Moldanubian Zone; compare Fig. 1.

At the transition between the NPZ to the MGCH, a significant clockwise rotation of $S_{\mathrm{Hmax}}$ can be observed within the MGCH (Fig. 11). A much lower but similar stress rotation can be seen at the transition from NPZ to MGCH in Fig. 1 along the river Rhine. Following that line, crossing the border to the STZ displays a counter-clockwise stress rotation, which is also suggested by the model (Fig. 11). Counter-clockwise rotation is also visible at the same unit boundary near the Main river. In general, the stress orientation in the STZ is counter-clockwise rotated to the orientations from the MGCH unit. $S_{\mathrm{Hmax}}$ orientation from the Harz (Fig. 1) seems to fit quite well to the model results (Fig. 11). However, structures there are complex and other factors could play an important role. Also taking the counter-clockwise rotation of $S_{\mathrm{Hmax}}$ from STZ to the MZ fits with modelling results. But the stress pattern in the molasses basins is probably more governed by the Alpine orogeny. In general, there are some similarities to observe, but frankly speaking, the model results are not able to prove the significant influence of the material properties on the stress orientation for this region.

Localization of individual grabens and volcanism in the European Cenozoic graben system can be related to late Hercynian fracture systems (Ziegler, 1992). The fan shaped stress pattern in the eastern part of the North German Basin has been explained as an effect of the close boundary to the stiff Eastern European Craton along the north-west to south-east striking Teisseyre-Tornquist Zone (TTZ, Goes et al., 2000; Gölke and Coblentz, 1996; Grünthal and Stromeyer, 1986, 1992, 1994; Kaiser et al., 2005; Marotta et al., 2002). This agrees well with the results of the models, where $S_{\mathrm{Hmax}}$ becomes perpendicular in a soft unit (NGB) directed to a stiff region like the East European Craton (e.g. model eee in Fig. 7).

Large spatial stress rotations are observed in Australia (Heidbach et al., 2018) and Northern America (Lund Snee and Zoback, 2018, 2020; Reiter et al., 2014). The variable basement structures there and consequently variable mechanical properties are good candidates to explain the complex stress pattern.




## 7 Conclusions

The impact of elastic material parameters (Young's modulus and Poisson's ratio), the body force (density) and low frictions discontinuities on the map view stress pattern is investigated. Each property is tested separately to avoid interdependencies.

505 This is realized with generic 3-D models using the finite element method. Within the models, three units with variable material properties are incorporated, where the boundary conditions govern the overall $S_{\text{Hmax}}$ orientation. The variation of density and the Poisson's ratio lead to small rotation ($\leqq 17°$) of the maximum horizontal stress ($S_{\text{Hmax}}$). In contrast to that, a stiffness contrast is able to produce significant stress rotation of 31° to 78°. Therefore, the variation of the Young's modulus in the upper crust are a potent explanation for observed stress rotation. Faults are represented in the models by cohesionless contact

510 surfaces. Observed far field stress rotation due to low friction faults ($\mu = 0.1$) is less as 3°. Implementation of low friction discontinuities in models with the Young's modulus anomaly leads to much smaller $S_{\text{Hmax}}$ rotation, in the order of 8° to 32°. Following that, faults did not produce far field stress rotation, they rather compensate stress rotation which are effect by Young's modulus anomaly. Comparison of model results with observed stress orientation in the region, which inspired the models, provides limited consistency.

515 *Acknowledgements.* Maps and model illustrations were generated using GMT software (Wessel et al., 2013). The author declare that he has no conflict of interest.





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
