# Peer review of "Stress rotation — Impact and interaction of rock stiffness and faults"

_Solid Earth, 2020_

## Referee Comment (RC1) · Anonymous Referee #1 · 31 Aug 2020

Stress rotation - The impact and interaction of rock stiffness and faults

By Karsten Reiter

The paper addresses stress rotation based on geomechanical numerical models. The models consist of different elastic units that are oblique to the direction of contraction. These units have variable mechanical properties and are separated by slipping surfaces. The geometry and boundary conditions are inspired by the geology of the German Central Uplands. A series of models are run based on a 3D finite element method. Results are presented to discuss the roles of density contrast, a variation of Poisson's ratio, a contrast of Young's modulus and friction of the discontinuities.

The paper is of interest in the context of regional studies in central Europe, for applied

studies dealing with local or regional stresses, and more generally for better under-standing stress field at the regional scale in inhomogeneous areas. The results pre-sented in the paper are interesting. However, I found that, while the literature review sections are lengthy, the methodology and the result sections are a bit sparse and need further clarification/explanation. I also found the paper difficult to follow and both the writing and the organisation of the paper need to be improved before publication. I recommend the publication after a major revision. Specific and technical comments are detailed below.

Specific comments:

Introduction and Section 2. The author decided to provide a short introduction followed by a section of literature review (section 2). I am not sure that this separation is ap-propriate for this paper. Section 2 is a bit lengthy, difficult to read and has a significant amount of information that I think is not needed in the frame of this manuscript. I think that deleting unnecessary information and merging contents of section 2 with the intro-duction and with section 4 will improve the manuscript. See below some suggestions for shortening the text.

L.54-61: I will delete this because it is detailed after when introducing the three orders.

L.62-69. These sentences could be significantly reduced. For example: "There are several features in the continental crust that can modify stress pattern on a local or regional scale. These features can be classified depending on their spatial coverage (Heidbach et al., 2007, 2010, 2018; Zoback et al., 1989; Zoback, 1992). According to this classification, stress sources refer to as first, second and third-order extend over distances >500 Km, between 100–500km and <100km. These distances are larger, approximatively the same, and smaller than the thickness of the lithosphere, respectively."

I don't think that it is necessary to introduce what is a stress. A reference to one textbook existing on the topic should be enough. Therefore, from my point of view,

section 2.2 can be entirely removed. Stress parameters such as Sh, SH and SV can be introduced when they are first encountered in the manuscript.

I don't think that section 2.3 is relevant. I think that referencing the world stress map in the geological setting is enough.

L.124-135. This paragraph is difficult to read and could be simplified as follow: "This study focuses on stress rotations that occur horizontally, i.e. in the map view. A vertical uniform stress field is assumed, which is consistent with previous studies (Heidbach et al., 2018; Zoback et al., 1989; Zoback, 1992). Vertical stress rotations observed within deep wells (Schoenball and Davatzes, 2017; Zakharova and Goldberg, 2014), due to evaporites (e.g. Cornet and Röckel, 2012; Röckel and Lempp, 2003; Roth and Fleckenstein, 2001), or man-made activities in the underground (e.g. Martínez-Garzón et al., 2013; 135 Müller et al., 2018; Ziegler et al., 2017) are not considered for simplification. On a map view, several potential sources of stress can superpose on top of each other and the resulting stress at a certain point comprises the sum of all stress sources from plate wide to very local stress sources. Differences between the resulting stress orientation and the regional stress source can be described by the angle $\gamma$ (Sonder, 1990), which can be substantial and can last in a change of the stress regime (Jaeger et al., 2007; Sonder, 1990; Zoback, 1992). ". But I am not sure what is the meaning of "can last in a change of the stress regime".

L.152-158: "Mechanical strength describes the material behaviour under the influence of stress and strain." I am not sure about this definition. From my understanding, rock strength refers to the capacity of the rock to fail but not to the elasticity.

L.167-170: "Small differences between the horizontal stresses increases the effect of faults on the local stress pattern, whereas large stress differences lead to more homogeneous stress pattern" This is not clear. This depends on the orientation or the fault relative to the stress and this depends on the difference between the max and min principal stresses S1 and S3, which are not always Sh and SH.

Section 3. Alike section 2, this section is difficult to read.

L.175-180. "These data. . . Anatolia" I think that these sentences could be removed.

L.195-200: "Among other things. . . 2006". This is not clear. You could replace by "In particular, these previous studies investigated the impact of. . ...

Section 4. I have several questions concerning the methods and assumptions. Some of these questions are partly addressed in the discussion, but I think the author should provide further justifications and clarifications. This does not necessarily imply running additional models, but the author should clarify, explain and justify the limitations of the models.

The dip of the contacts between the units (vertical) should be indicated in the methodology. How is it compared to the dip of the structures in Germany? How does the dip used in the models impact the results?

I found the term "basic material" not very clear, maybe use "reference material".

Maybe the author can provide some illustrations of the actual model and mesh. The materials are elastic, but what about its strength (capacity to fail, see previous comment L.152-158)? Is it possible that the materials reach failure in some parts of the models due to the boundary conditions and stress concentrations? How this could modify the results? More generally, there is no indication of the stress magnitude within the model, except for the boundary condition.

I am afraid I do not fully understand the boundary conditions. What is the pre-stressed basic model, initial stress? Where is the virtual well? For which model does the boundary condition is calibrated? I imagine that different boundary conditions will be needed to fit the stress profile presented in Fig.4 depending on the rock properties. Is the boundary condition similar for all models? What is the impact of the boundary condition on the results? I agree that SHmax is more difficult to calibrate, however, it seems that the chosen value is significantly different than the one provided in the Brudy et al.

(1997). In particular, there is no change in the stress regime in the case of Brudy et al. (1997). I think it is important to understand the impact of this change in the stress regime in the models and discuss it, as it is poorly constrained.

The results concern the stress orientation at a depth of 1000 m below the surface, where there is a strike-slip fault regime according to the chosen boundary condition. How do the results change with depth and as a function of the stress regime? Why the results only concern the depth orientation at 1000 m? Also, do the rotations only occur in map view or is there also stress rotation in cross-section. In other terms, is the plan of observation presented comprise the principal stress?

The author tests separately different parameters: the density, the PR and the YM. The ranges of parameters tested seem correct. Models are designed to test each parameter individually. I think that this is a relevant method for a generic study. I am wondering however what is the geological meaning of this. In nature, these parameters can be interdependent. For example, a rock with a low density may have a low YM as well. Also, do the materials and discontinuities have constant properties with depth? How does this potentially impact the results of the models? I recommend the author to discuss further these points.

Section 5.

In the methodology, it is stated that the model is orientated relative to the North. Therefore, why no presenting the results with actual stress orientations rather than talking about clockwise and anti-clockwise rotation. Maybe this will help a little bit the reading.

I found the results difficult to compare between the models. Maybe a cross-plot or a histogram comparing the various models could help.

I found that the results section lack of explanations of the behaviour of the model. For examples, what causes the rotation in the case of the density models? Why the models with the faults have little rotations. Is it because the faults are not critically

stressed because they are not optimally orientated? More generally, I encourage the author to provide rationals for the observed behaviour.

Section 6.

It seems that the author made a significant effort in reviewing previous studies on stress rotations. I will suggest providing a table summarizing this. This will help the reader and be an added value for the manuscript.

Concerning stress rotation and faults. Do we expect the stress rotations to be time dependant and change between when the fault is locked and when the fault slip? Concerning the results from the upper panel in Fig. 10. There is a significant difference between the behaviour in the shallow part of the model and the deeper part. Is this related to the boundary condition and the fact that the faults are critical stress in the strike-slip regime and locked in the normal faulting regime?

I think that section 6.8 should be presented in the result section and not in the discussion. "The comparison will concern only the results of the models investigating the Young's modulus, as this material parameter have the strongest impact" but the model integrate densities and Poisson ratio according to table 2? Also, to help the reader, maybe the author could cross-plot the stress rotations from the model with the stress rotations from Fig. 1.

General organisation. Generally, the necessary pieces of information are provided in the manuscript, but I found that the paper lack clarity and organisation. The investigated geometry is interesting. I think however that the results depend on the chosen geometry, especially the strike and dip of the discontinuities/units and are therefore limited in scope. From my point of view, this work is both a generic study and a simplified study case. Accordingly, I recommend the author to re-organise their manuscript into a more classic scheme. For example (1) an introduction that merges the content of sections 1 and 2 after removing all unnecessary materials, with a review of stress rotation, a review of the key controls and a summary of the objectives (a generic study that

aims to identify the key parameters and a case study in central Europe where there is good coverage of the stress field and a good knowledge of the regional geology to test the parameters). (2) Methodology. (3) Geological setting of the study case. (4) Result section divided into (i) generic models and (ii) a more realistic model. (4) Discussions centred on comparing the results with previous works.

Text. I think that the manuscript suffers from numerous English mistakes and unclear sentences. This sometimes obfuscates the message and I think that the manuscript, in general, will benefit from thoroughly polishing the text before publication. Several suggestions are provided below, but this is by no means an exhaustive list.

Technical comments:

L.31: "It was suggested, that" remove coma after suggested.

L.35: "sediments, and were" remove "and".

L:37-40: These sentences are not very clear. I am not sure to understand what "the assumed stress pattern (stress rotations)" refers to.

L.39: "can only partly explained" replace by " can only be partly explained".

L.43: "These 2-D models was" replace by "These 2-D models were".

L.43: "stress pattern" replace by "stress patterns".

L:43 ", applying" replace " by applying" and remove coma?

L.44-46: Maybe this could be simplified. For example: However, these 2D models cannot account for topography, crustal thickness and depth variability in stiffness and can overestimate horizontal stresses (Ghosh et al., 2006). Furthermore, none of these previous studies compared the impact of the influencing factors separately.

L.47-48: Not clear. In this paper, we use a series of generic models to identify which properties can cause substantial stress rotations away from a material transition or a

discontinuity.

L.49: "orientations" replace by orientation.

L.49: "north-south orientation" N-S orientation.

L.51: Orientations are usually given with three digits. N030°.

L.50-51: "The basement structures there are striking about 30âŮę , which is perpendicular to the observed SHmax orientation" actually N030° is not perpendicular to N150°, there is a 20° misfit.

L.57: "The second major driver are" replace by "The second major driver is".

L.57-58: "Plate boundary forces where identified and derive deviatoric stresses" this sentence is not clear.

L.62 : "The most of these features" replace by "Most of these features".

L.62-135: see some corrections in the Specific comments section.

L.135: "both is not a subject of that study" replace by "both are not the subject of this study".

L.168: "between the horizontal stresses increases" remove s at increases.

L.169: " is investigated" replace by "has been investigated"

L.191-192: "The explanation of that crustal structure are a cold, dense and slowly subsiding lithospheric root beneath the Alps (Müller and Zürich, 1984)." replace by "This crustal structure can be explained by..."

L. 193: "an subject" replace by "the subject"

L.195: "which factors contributes" remove s at contributes.

L.207: "topography has major effects" replace by "topography have major effects"

L.232: "grade varies considerably" remove s at varies.

L.243: "Model dimension" maybe replace by "Model geometry".

L.247: "oriented 30âŮę counter-clockwise from east-west" why not just say oriented N030°.

L.248: "three unit" replace by 'three units".

L.257-260: "The finite.... 1986)." I think that the contents of these sentences are unnecessary. Maybe replace by " The stress orientations in the models are investigated using the finite element method (FEM)".

L.274: "Figure 3 visualize" replace by "Figure 3 visualizes".

L.277: "Young's modules" replace by "Young's modulus".

L.279-280: replace "where" by whereas" and "mantel' by "mantle".

L.314: "result in" replace by "results in a slightly counter-clockwise".

L.341: "degreases" replace by "decreases".

L.343: "exhibits" and "displays" remove s.

L.345: In terms of mechanical properties, the opposite of "stiff" is usually "compliant" (elasticity), whereas "weak" is the opposite of "strong" (strength).

L.368: "Fore sure it is really unlikely that" replace by "It is unlikely that".

L.376: "in a depth of 1000 m depth" replace by "at a depth of 1000 m".

L.388: " are may be" remove are.

L.390: remove coma after suggest.

L.392: this sentence is not clear.

L.395: "the relative rotation are" replace by "the relative rotations are".

L.401-402: " The importance of stiffness differences is an result of other models too.." This sentence is not clear. "Similar impacts of stiffness contrast have been described in previous works. . ."

L.415-417: These sentences are unclear.

L.424: "this generic models" replace by "these generic models".

L.462: There is 6.8.1 but no 6.8.2?

L.430: "indicates" replace by "indicate"

L.431: The work by Petit and Mattauer, (1995) concern mesoscale faults and I am not sure about this 2 km distance indicated here.

L.431: "is to" replace be "can be"

L.466: "parameter have the" replace by "parameter has the"

L.477: "should deflected" replace by "should be deflected"

I hope this will help to improve the manuscript.

---

## Referee Comment (RC2) · Anonymous Referee #2 · 12 Oct 2020

General comments

The article addresses an interesting problem of tectonic stress deviation due to the contrast in mechanical properties and fault motion. To explore this problem the Author designed models built of elastic and contact elements, loaded with body forces and tectonic strain. The models have a very simple structure and seem to be correctly constructed and solved. Some of the obtained results are interesting. However, this study, as well as the text itself, has many significant drawbacks, which I will focus on below. I do not mention language issues as I have no competence in this area.

Critical remarks:

Chapters 1. 2. and 3. Introduction

[Figure]

The Introduction is very long and exhausting, resembles an academic lecture. There are summarized various aspects of stress generation and measurement, very loosely related to the research subject. In my opinion, such chapters as 2.1, 2.2, 2.3 are not necessary and should be altogether shortened to one paragraph. On the other hand, the geodynamic context of stress rotation is poorly introduced. It would be better to present several natural examples of stress rotation and their possible reasons. The passages related to stress rotation in mechanical models can also be extended. Also chapters: 3.1. and 3.2 should be shortened significantly, as the European regional context, as it turns out at the end of the article, does not play an important role in the evaluation of the modeling results. Instead, more patients should be paid to the realistic crust profiles which determine ranges of material properties like density, stiffness, and to the effective friction coefficient of regional fault zones. I would suggest focusing on the key factors important for this modeling study.

Chapter 4. Model Setup

The structure of the model and boundary conditions are not sufficiently described. E.g. it is not clear how the faults terminate at the model boundary and what is a dip of them. In my opinion, the definition of constant elastic properties and density for the entire crustal thickness based on the property of rock present in the upper crust is a bad idea. Among the lithologies presented in Fig. 3 only granite can build the entire crust. Even the simple models should consider a realistic range of material properties, evaluated from typical lithologies of the Earth's crust. Neither in the introduction nor in the model setting chapter there is no reference to the realistic lithological profile of the Alpine foreland plate, based on geophysical constraints. Some references are in the discussion but without references to the lithological composition of the crust. The densities are given in Table 2 like 2.1. - 2.2 g/cm3 are typical for salt rock but for the crystalline crust !. The constant density across 30 km thick crust is also unrealistic. The results of modeling could be more significant when realistic and geophysically documented crust properties were used. The author should also justify such a dramatic

change in YM across the entire crust. I can imagine that tenfold differentiation of effective stiffness can be produced by e.g. high heat flux variations, which are not present in the reference units. However, the assumption of elastic crust in the area with a moderate surface heat flow density is difficult to defend. The crust is probably rheologically layered, thus the larger part of the weaker crust unit is inelastic. Even accepting elastic simplification of the model the stiffness and structure of the model should follow from the realistic lithological and rheological profile of the crust.

Chapter 5. Results of modeling

The results of modeling for variable PR, and density, even for unrealistically high contrast of parameters did not give significant results. Also, the stress rotations at faults are negligible, which is probably caused by their orientation under a high angle to SHmax which is not preferential for reactivation and by their ideal planar geometry. The normal fault stress regime in the lower part of the model (mentioned by the author before) is additionally the reason why there is vanishing shear stress at the vertical plane of the fault. However, the understanding of modeling results needs a better explanation of the fault implementation in the previous chapter. More significant but quite obvious results were obtained for YM variation, although the contrasts in this parameter are unrealistically high. The more significant modeling results require closer examination. It would be good to check the sensitivity of the model to changes in YM and present it on graphs or in the table. The most interesting results were obtained for combined Young modulus and fault slip. The results point that stress rotations between units of contrasting stiffness can be reduced by active faults. However in this case the analysis and presentation of fault displacement are necessary. When exploring this on realistic parameters and geometries the paper could be more interesting.

Chapter 6. Discussion

In chapter 6.1, the Author honestly states that the adopted assumptions regarding the material parameters do not match the model of the Earth's crust. The elastic thickness

fits more closely with the lithosphere than the crust. So the Author is aware of the weaknesses of these models but why not translate it into a realistic model setup. In this chapter the result of stress changes with depth are presented, which better match chapter 5. Such analysis could be interesting, but unrealistic material properties and especially pure elastic mechanics make them not applicable to the Earth's crust. The interesting results of experiments with coupled faults and YM changes could be better presented and the factors governing regularities in obtained results should be much deeper explored and explained.

Chapter 7. Conclusions

In conclusion, the author admits that the modeling results do not reflect the stress rotation in the reference part of the Alpean foreland. Conclusions are very modest in comparison to the volume of the paper.

Some detailed remarks.

135 "Density contrast and topography" The topographic stress is a separate and wide subject, not investigated in this study. It is better to skip this issue instead of just put a number of references. 140 " stresses due to topography and crustal inhomogeneities are in the order of tens of MPa,140 which are on a similar magnitude as the plate boundary forces". In this case, stresses should not be directly compared to forces. 153 "Mechanical strength describes the material behaviour under the influence of stress and strain. The focus here is on elastic material properties, characterized by the Young's modulus and the Poisson's ratio." Please consider that elastic properties do not characterize strength. 168 "Small differences between the horizontal stresses increases the effect of faults on the local stress pattern" To some extent, because low differential stress means also low shear stress at the fault plane. 368 "Fore sure it is really unlikely that such constant materials with such a thickness exists somewhere in the crust" YES for sure. Then why such unrealistic materials were tested? 371 "However, the overall geometry seems reasonable, as the brittle domain or elastic thickness of the

crust (Te), which is a measure of the integrated strength of the lithosphere" Please consider that this is a lithosphere but not the crust itself. The upper mantle often contribute to this elasticity, then the mechanical properties assumed for the model are even less appropriate. 410 "The observed radial stress pattern southward of the Bohemian massif (Reinecker and Lenhardt, 1999) agrees 410 well with this study" As there is a lack of good examples of data constraining presented modeling results, this special case could be illustrated in the figure. 455 "The observed stress rotation strongly depends on the depth. In the soft units, SHmax rotates counter-clockwise near the surface (0 to -8 km). In contrast to that a clockwise rotation can be observed in greater depth (18−30 km)" Such an interesting result can be presented in more detail in Chapter 5. However, only the results from the upper part of the models are significant, due to the inelastic effects prevailing in the lower part. 491 "but frankly speaking, the model results are not able to prove the significant influence of the material properties on the stress orientation for this region." That means, that the area selected for model evaluation is incorrect.

To summarize, the paper presents simple and unrealistic models of the Earth's crust. A large part of the text is too long or unnecessary, while the most interesting points are insufficiently described. I would recommend a major revision or rejection of this paper.

---

## Short Comment (SC1) · 29 Oct 2020

I agree with a number of issues that the reviewer raises, such as length of the intro, more details on the model setup and a better reasoning why the material properties are simplified, i.e. homogenous with depth. However, I think we can learn a lot from a simple model and maybe can derive hypothesis why we observe rotation of the maximum horizontal stress SHmax on scales of 50-200 km in intraplate areas where little topography is present (USA: Lund and Zoback, 2020, Nature Communications, doi:10.1038/s41467-020-15841-5 or Australia: Rajabi et al., (2017) Earth Science Reviews, doi:10.1016/j.earscirev.2017.04.003). Putting more complexity into the model has the risk that it may produce ambiguities in the interpretation what causes the observed rotation of the stress tensor. Thus, the generic study of Reiter can help indeed

to better understand potential sources of these somehow unexpected rotations.

However, as most of the data of the SHmax orientation in intraplate settings are within the upper 15 km (for the above mentioned USA and Australia examples data are to large extend in the upper 5 km from borehole logs) I would focus on the upper elastic part of the crust, i.e. the upper 10 km - than the elastic approach is a reasonable justification.

Another issue raised is the missing variability of the rock properties with depth. I agree that this "feels" like an oversimplification, but the SHmax orientation does not change significantly with depth except where mechanical decoupling due to e.g. evaporate layers occurs. But this has to be explained in more detail and maybe a sensitivity test could show that the key findings are not affected by this simplification.

Thus, I agree that the model is simple, but it is not unrealistic. The question is if the simplifications are justified and sufficient to address the key question that the model is investigating. And to quote George Box' aphorism "All models are wrong, but some are useful" the question in the very end is if this model setup is helpful. The answer from my point of view is yes (after the author shows or better describes that the simplifications do not affect the key findings).

---

## Author Comment (AC1) · 1 Nov 2020

I have to thank Referee #2 for having read and reviewed the manuscript carefully and in detail. The reviewer correctly summarises, that the manuscript "... presents simple and unrealistic models of the Earth's crust". Realistic models with a complex geometry, variable material properties, laterally and with depth, and a complex rheology are appropriate to reproduce observed mechanical features in detail. This is suitable for generating best-fit-models. However, the more 'adjusting screws' are technically implemented, the easier it becomes to achieve an optimal fit to the observation. To me the question is: Are such best-fit-models suitable for identifying the most important parameters for stress rotation in the crust? Do such complex models give us a better understanding of the interaction of the properties used and discontinuities, which

create or prevents stress rotation? For me, the answer is simple: No.

It has never been the aim of this study to present a model of the earth's crust that is as realistic as possible. It was mainly concerned with identifying the influence of density, elastic material properties (Young's modulus and Poisson's ratio) and discontinuities on the stress orientation, which deviates from the assumed stress orientation due to plate boundary forces. Therefore, simple generic models are used, to test each parameter separately at first. Interaction of these parameters tested afterward.

Referee #2 mentions the following regarding the models on which the mechanical properties of the Variscan units are tested: "This means, that the area selected for the model evaluation is incorrect". This area has inspired the model geometry. A reader of the manuscript would never understand the chosen geometry, neglecting that background of observation (stress orientation) and zoning in the Central German Highlands. The model, which uses the variation of material properties from the German Variscides, reproduces some observed stress orientation pattern, and some not. Therefore, I summarized carefully: ". . . the model results are not able to prove the significant influence of the material properties on the stress orientation for this region." This is true, because I could not exclude other factors that contribute significantly to the observed stress rotation. Furthermore, a model is never able to prove a hypothesis. The statement in my conclusion "Comparison of model results with observed stress orientation in the region, which inspired the models, provides limited consistency" is in contrast to the re-interpretation of Referees #2: "In conclusion, the author admits that the modeling results do not reflect the stress rotation in the reference part of the Alpean foreland."

Referee #2 suggest presenting more details of the model results, such as different stress rotation with depth, or more details of the models with variable Young's modulus, separated by low friction faults. However, I am aware of the simplicity of the models used. Therefore, I have avoided to present too much details, which probably would pretend the use of realistic scenarios. Much more realistic models are needed to study such details. According to Samual Karlin's remark: "The purpose of models is not to fit

the data but to sharpen the question.", the used simple generic models sharpened the question for detailed studies in the future.

The major findings of that study, that (1) the Young's modulus is capable to cause significant stress rotation in the crust away from the material transition, (2) low friction discontinuities do not cause significant stress rotation away from that structure, and (3) low friction faults are capable to compensate stress rotation due to different Young's modulus are independent from the question, up to what depth in the crust, elastic material properties are sufficient. I think this interesting question is a fundamental one that cannot be adequately addressed in this study.

---

## Author Comment (AC2) · 24 Nov 2020

I have to thank Referee #1 for having read and reviewed the manuscript carefully and in detail. She/He understood both, the indention of the paper and the modelling approach (generic models) very well. RC1 gives many detailed suggestions to shorten and clarify the manuscript, which are very helpful. RC1 proposes to reorganize the paper by integrating the section "Stress in the Earth's Crust" into the introduction in a shortened form. That's a reasonable suggestion.

In the following, I will give comments and answers to specific questions or suggestions. RC1's questions are marked by quotes.

"The dip of the contacts between the units (vertical) should be indicated in the method-

ology. How is it compared to the dip of the structures in Germany? How does the dip used in the models impact the results?"

All contacts are vertical in the used models. For the zone transitions in the German Variscites, NNW directed trusting combined with sinistral but also dextral shearing due to oblique convergence has been identified. The dip angle of that units is uncertain, but according to some authors probably only about 10°. Furthermore, changes of rock type and material properties with depths are uncertain. Only reflection seismic profiles (DEKORP) from the 1980's and their interpretations are available. A variable dip angle like the suggested one has not been tested. I don't want to speculate on potential results, but a resulting pattern will be different to the presented model results. However, it was not the goal to model that region appropriate. The focus was on the question to identify the impact of the chosen material properties, low friction discontinuities and their interaction as well.

"Maybe the author can provide some illustrations of the actual model and mesh. The materials are elastic, but what about its strength (capacity to fail,. . . )? Is it possible that the materials reach failure in some parts of the models due to the boundary conditions and stress concentrations? How this could modify the results? More generally, there is no indication of the stress magnitude within the model, except for the boundary condition."

The visualization of the mesh would need a big figure, as the model even in the map view (x-y-direction) consists of about 11,000 Elements. The lateral resolution is 3km, as described in the manuscript. As only elastic material properties are used, failure is not possible. I tested the EEE and eee Model using two different Coulomb failure criteria. The models are run first with a cohesion (C) of 30 MPa and a friction angle (FA) of 40° and second with C= 10 MPa and a FA of 30°. For the EEE model with C=30 MPa and FA=40°, no failure (Yield criteria <1) will be reached. For the eee model using that criteria and for both models (EEE and eee) using C=10 MPa and FA=30° failure occurs. Conditions of failure or close to failure (Yield criteria ~1 or >1) occurs

only near the surface (a few km) and close to the material transition (~20-30 km). Around the material transition, (near) failure can be observed within the stiff units only. For the EEE model with C=10 MPa and FA=30°, failure is more spaciously distributed near the surface. In the case of failure, the SHmax (SH) orientation will be balanced, which means, that SH rotates back in the north-south orientation, like the boundary conditions. In general, failure compensates stress rotation in the same way like low friction contact surfaces (faults).

The stress magnitudes of the reference model are shown in figure 4. Magnitudes of the other models are similar on the same place, depending on the material properties and whether there is a significant material transition in the neighbouring unit or not. For models having a lower Young's modulus, Shmin (Sh) and SH are lower, and the opposite for the models with the larger Young's modulus. Usage of variable material properties within the model units lead to different stress magnitudes within these units. The adjustment of the boundary conditions would always depend on which point of comparison would be chosen. Overall, the boundary conditions are defined by displacement, based on the reference model.

"I am afraid I do not fully understand the boundary conditions. What is the pre-stressed basic model, initial stress? Where is the virtual well? For which model does the boundary condition is calibrated? I imagine that different boundary conditions will be needed to fit the stress profile presented in Fig.4 depending on the rock properties. Is the boundary condition similar for all models? What is the impact of the boundary condition on the results?"

Before boundary conditions will be applied to the model, initial stress conditions are needed. These initial stress condition are a more complex issue. First of all, gravity acts instantaneous to a model, which would lead to significant settlement of the model. The initial stress condition avoids the settlement to a large extend. Furthermore, the observed ratio between SV (vertical stress)) and SH plus Sh are not a linear function with depth in contrast to simple isotropic or hydrostatic assumptions. The k-ratio

((SH+Sh)/2*SV) is much larger than one in the first hundred to thousand(s) of meters and becomes smaller than one in greater depth. This observation is supported by many data, but also by a semi-empirical function, provided by Sheorey(1994). His model is representative for an Earth without topography and tectonics. By settling the model without lateral shortening, but a large Poisson' ration under gravitational load, the derived stress state provides a quite good fit to Sheorey's profile as a function of depth and the Young's modulus. Therefore, the initial stress conditions provide a stress state with depth, which accounts broadly speaking for conditions without tectonic shortening. The models are pre-stressed in this way and underwent the tectonic component, or the difference between Sh and SH by the application of lateral boundary conditions. The subject of initial stress condition where kicked out, because reviewers usually are not interested in that subject and asks to skip that topic.

The boundary conditions are simple, the model is shortened in north-south direction by 400 m and extended by 60 m in east-west direction, as shown in Fig 2. of the manuscript. The virtual well is in the centre of the model. The boundary conditions are calibrated on the reference model (Fig 4.). These boundary conditions are applied to all the different model runs. The usage of a different Young's modulus leads to a change of the magnitudes of SH and Sh. A larger Young's modulus will lead in larger stress magnitudes and vice versa. However, as mentioned earlier, a fit of a virtual well (stress profile) from a heterogeneous to a homogeneous model depends on the selected area or unit. Therefore, homogeneous boundary conditions are chosen. Less shortening/extension along the boundaries would lead to a lower difference between SH and Sh.

"I agree that SHmax is more difficult to calibrate, however, it seems that the chosen value is significantly different than the one provided in the Brudy et al. (1997). In particular, there is no change in the stress regime in the case of Brudy et al. (1997). I think it is important to understand the impact of this change in the stress regime in the models and discuss it, as it is poorly constrained."

It depends on the chosen way to put a regression line on Brudy's SH data. There is a change of rock properties with depth at the KTB site, which is visible in the magnitude data. I wouldn't take too much effort to fit the model to the SH magnitudes. They are not measured; they are calculated based on several assumptions. The rocks in greater depths, like the KTB well becomes more and more ductile. That was one of the reasons, why they had to stop drilling. To me, a proportional increase of SH versus Sh over lager depth sections is questionable. From the modelling standpoint, it is hard to model such increasing differential stresses. Complex boundary conditions are needed combined with low Poisson's ratio's for greater depths.

A change of the stress regime from TF to SS and to NF in greater depth like visualized by the virtual well is assumed by several authors like for the Alberta basin. Similar changes of the stress regime are reported for many other regions. However, the change of the stress regime with depth is not a subject of that manuscript. The stress orientation is visualized for a depth of 1000 m below surface. This depth is not affected by general changes of the stress regime from one of the models to another one. More important, the stress magnitudes become smaller within the stiff units next to the soft units. This seems to be the major reason for the observed stress rotation (and observed reaching of failure, using a Coulomb criteria), not the change of the stress regime.

"The results concern the stress orientation at a depth of 1000 m below the surface, where there is a strike-slip fault regime according to the chosen boundary condition. How do the results change with depth and as a function of the stress regime? Why the results only concern the depth orientation at 1000 m? Also, do the rotations only occur in map view or is there also stress rotation in cross-section. In other terms, is the plan of observation presented comprise the principal stress?"

The orientation of SH changes with depth, which is shown in Fig 10. The visualization of SH orientation is always in a depth of -1000m, to make it comparable. The scope of that paper is to investigate the impact of the material properties and faults,

as mentioned in the introduction. Due to the simplification of the model, I don't want to interpret the stress changes with depth too much in detail. However, the stress rotation decreases with depth (Fig. 10). Next to the material transition, SH and Sh becomes smaller near the surface and larger near the bottom of the model, within the stiffer units. In other words, the units with a lower Young's modulus causes a reduction of the stress magnitudes (SH and Sh) within the neighbouring stiffer units near the surface and increasing stress magnitudes in greater depth. It is not an issue of regime change. Magnitudes of SV, SH and Sh are pretty close to the principal stress magnitudes.

"The author tests separately different parameters: the density, the PR and the YM. The ranges of parameters tested seem correct. Models are designed to test each parameter individually. I think that this is a relevant method for a generic study. I am wondering however what is the geological meaning of this. In nature, these parameters can be interdependent. For example, a rock with a low density may have a low YM as well. Also, do the materials and discontinuities have constant properties with depth? How does this potentially impact the results of the models? I recommend the author to discuss further these points."

Sure, the tested material properties are not independent in nature. However, there is no linear relation between density, Young's modulus nor the Poisson's ratio. As shown by Fig. 3, density and Young's modulus for sediments are quite variable. For example, a limestone can have a smaller density, but larger Young's modulus as a shale. However, there is no geological meaning of the material variation, as it is a generic study. Sure, these properties will change with depth (or temperature) This is not included in the model. But usage of linearly increasing rock properties will affect the model results only slightly. Important is the effect of the material transition. For linear increasing density, Young's modulus or Poisson's ratio, the resulting effect remains similar.

"... what causes the rotation in the case of the density models? Why the models with the faults have little rotations. Is it because the faults are not critically stressed because they are not optimally orientated? More generally, I encourage the author to provide

rationals for the observed behaviour."

Gravitational load is one of the major sources of stress in the Earth crust. This body force effects the magnitude of SV. However, due to the mechanical effect, which is specified by the Poisson's ratio, a locally increased SV lead to a local increase of SH and Sh, which can affect the stress pattern locally too.

The models with the faults have SH rotation next to the faults, as the principal stresses are always parallel or perpendicular to a fault (contact) like next to a free surface. However, observed stress orientation, which are presented in the figures are always placed away from the faults (>12.5 km), as I am only interested in the far field effects, not the near field effects. This near field effects around faults would be a different subject. The orientation is not optimal, but the chosen properties (C=0, FA∼5°) are that low, that the units on each side are decoupled from each other for sure. In general, active faults do not have a large impact on the far field stress pattern at all, except close (∼<10 km) to the discontinuity. That distance again depends on the Young's modulus.

"Concerning stress rotation and faults. Do we expect the stress rotations to be time dependant and change between when the fault is locked and when the fault slip? Concerning the results from the upper panel in Fig. 10. There is a significant difference between the behaviour in the shallow part of the model and the deeper part. Is this related to the boundary condition and the fact that the faults are critical stress in the strike-slip regime and locked in the normal faulting regime?"

In the case of homogenous material properties, the far field stress pattern is independent whether a fault is locked or not, as shown by the model series with variable friction properties. The stress rotation is not time depended for the models, it's a function of the variable material properties and the boundary conditions. The faults in the models did not have any cohesion, the friction coefficient is quite low (0,1) which is a friction angle of about 5-6°. The different orientation of SH in the shallow and deeper parts are not affected by the boundary conditions.

The complex pattern in the upper cross-section (eEe) of Fig. 10 is a product of the fact, that no fault (or a locked fault) is included at the material transition. The pattern is a product of the interaction of the different Young's modulus in the neighbouring units. This leads to a reduction of SH and Sh near the surface, next the material transition within the stiff units. This leads to the complex stress pattern. Active faults decouple that (|e|E|e|, Fig. 10). They balance the stress orientation (and the stress magnitudes to a certain point).

"I think that section 6.8 should be presented in the result section and not in the discussion. "The comparison will concern only the results of the models investigating the Young's modulus, as this material parameter have the strongest impact" but the model integrate densities and Poisson ratio according to table 2? Also, to help the reader, maybe the author could cross-plot the stress rotations from the model with the stress rotations from Fig. 1."

Yes, it is an option, to place section 6.8 in the result section. Yes, the models apply all estimated material properties from the Variscan units in Germany (Table 2). It could be an option, to plot an average orientation from Fig 1 in comparison next to fig 11, allowing a better comparison.

---

## Author Response (AR1)

*Reviewer comments in italic and dark green*
My comments in black and normal letters
Changes are indicated in dark blue

**Reviewer 1 (https://doi.org/10.5194/se-2020-129-RC1, 2020)**

*The paper addresses stress rotation based on geomechanical numerical models. The models consist of different elastic units that are oblique to the direction of contraction. These units have variable mechanical properties and are separated by slipping surfaces. The geometry and boundary conditions are inspired by the geology of the German Central Uplands. A series of models are run based on a 3D finite element method. Results are presented to discuss the roles of density contrast, a variation of Poisson's ratio, a contrast of Young's modulus and friction of the discontinuities.*

*The paper is of interest in the context of regional studies in central Europe, for applied studies dealing with local or regional stresses, and more generally for better understanding stress field at the regional scale in inhomogeneous areas. The results presented in the paper are interesting. However, I found that, while the literature review sections are lengthy, the methodology and the result sections are a bit sparse and need further clarification/explanation. I also found the paper difficult to follow and both the writing and the organisation of the paper need to be improved before publication. I recommend the publication after a major revision. Specific and technical comments are detailed below.*

*Specific comments:*
*Introduction and Section 2. The author decided to provide a short introduction followed by a section of literature review (section 2). I am not sure that this separation is appropriate for this paper. Section 2 is a bit lengthy, difficult to read and has a significant amount of information that I think is not needed in the frame of this manuscript. I think that deleting unnecessary information and merging contents of section 2 with the introduction and with section 4 will improve the manuscript. See below some suggestions for shortening the text.*

KR: Several of the proposed changes and cuts were made in accordance with the suggestions, see below for details.

*L.54-61: I will delete this because it is detailed after when introducing the three orders.*

KR: This change was made according to the recommendation

*L.62-69. These sentences could be significantly reduced. For example: "There are several features in the continental crust that can modify stress pattern on a local or regional scale. These features can be classified depending on their spatial coverage (Heidbach et al., 2007, 2010, 2018; Zoback et al., 1989; Zoback, 1992). According to this classification, stress sources refer to as first, second and third-order extend over distances >500 Km, between 100–500km and <100km. These distances are larger, approximatively the same, and smaller than the thickness of the lithosphere, respectively."*

KR: This part has been removed now.

*I don't think that it is necessary to introduce what is a stress. A reference to one textbook existing on the topic should be enough. Therefore, from my point of view, section 2.2 can be entirely removed.*

*Stress parameters such as Sh, SH and SV can be introduced when they are first encountered in the manuscript.*

KR: Proposed changes are implemented

*I don't think that section 2.3 is relevant. I think that referencing the world stress map in the geological setting is enough.*

KR: Proposed changes are implemented.

*L.124-135. This paragraph is difficult to read and could be simplified as follow: "This study focuses on stress rotations that occur horizontally, i.e. in the map view. A vertical uniform stress field is assumed, which is consistent with previous studies (Heidbach et al., 2018; Zoback et al., 1989; Zoback, 1992). Vertical stress rotations observed within deep wells (Schoenball and Davatzes, 2017; Zakharova and Goldberg, 2014), due to evaporites (e.g. Cornet and Röckel, 2012; Röckel and Lempp, 2003; Roth and Fleckenstein, 2001), or man-made activities in the underground (e.g. MartínezGarzón et al., 2013; 135 Müller et al., 2018; Ziegler et al., 2017) are not considered for simplification. On a map view, several potential sources of stress can superpose on top of each other and the resulting stress at a certain point comprises the sum of all stress sources from plate wide to very local stress sources. Differences between the resulting stress orientation and the regional stress source can be described by the angle γ (Sonder, 1990), which can be substantial and can last in a change of the stress regime (Jaeger et al., 2007; Sonder, 1990; Zoback, 1992). ". But I am not sure what is the meaning of "can last in a change of the stress regime".*

KR: Proposed changes are implemented.

*L.152-158: "Mechanical strength describes the material behaviour under the influence of stress and strain." I am not sure about this definition. From my understanding, rock strength refers to the capacity of the rock to fail but not to the elasticity.*

KR: The term 'strength' have been changed to 'stiffness'.

*L.167-170: "Small differences between the horizontal stresses increases the effect of faults on the local stress pattern, whereas large stress differences lead to more homogeneous stress pattern" This is not clear. This depends on the orientation or the fault relative to the stress and this depends on the difference between the max and min principal stresses S1 and S3, which are not always Sh and SH.*

KR: This sentence is rewritten

Large stress differences lead to a more homogeneous stress pattern (Laubach et al. 1992; Yale, 2003), whereas low differential stresses allow a switch of the stress regime caused by faults. (line 109)

*Section 3. Alike section 2, this section is difficult to read.*

KR: Proposed changes are implemented.

*L.175-180. "These data: : : Anatolia" I think that these sentences could be removed.*

KR: Proposed removals have been realized

*L.195-200: "Among other things: : : 2006". This is not clear. You could replace by "In particular, these previous studies investigated the impact of: : :..*

KR: Proposed changes are implemented.

*Section 4. I have several questions concerning the methods and assumptions. Some of these questions are partly addressed in the discussion, but I think the author should provide further justifications and clarifications. This does not necessarily imply running additional models, but the author should clarify, explain and justify the limitations of the models.*
*The dip of the contacts between the units (vertical) should be indicated in the methodology. How is it compared to the dip of the structures in Germany? How does the dip used in the models impact the results?*

KR: All contacts are vertical in the used models. For the zone transitions in the German Variscites, NNW directed trusting combined with sinistral but also dextral shearing due to oblique convergence has been identified. The dip angle of that units is uncertain, but according to some authors probably only about 10°. Furthermore, changes of rock type and material properties with depths are uncertain. Only refraction seismic profiles (DEKORP) from the 1980's and their interpretations are available. A variable dip angle like the suggested one has not been tested. I don't want to speculate on potential results, but a resulting pattern will be different to the presented model results. However, it was not the goal to model that region appropriate. The focus was on the question, to identify the impact of the chosen material properties, low friction discontinuities and their interaction as well. The changed description make the used inclination of the contact more clear:

In the centre of the model, three diagonal units having a width of 50 km are oriented 60° from north. The unit boundaries are vertically incident. (line 181)

KR: Furthermore, the subject of dip within the geological structures are included within the discussion chapter now.

*I found the term "basic material" not very clear, maybe use "reference material".*

KR: The term 'basic material' have been changed to 'reference material'.

*Maybe the author can provide some illustrations of the actual model and mesh.*

KR: The visualization of the mesh would need a big figure, as the model even in the map view (x-y-direction) consists of about 11,000 Elements. The lateral resolution is 3km, as described in the manuscript.

*The materials are elastic, but what about its strength (capacity to fail, see previous comment L.152-158)? Is it possible that the materials reach failure in some parts of the models due to the boundary conditions and stress concentrations? How this could modify the results?*

KR: Within the discussion chapter, that subject is discussed now: see section 'Failure criteria' (line 433)

*More generally, there is no indication of the stress magnitude within the model, except for the boundary condition.*

KR: The stress magnitudes of the reference model are shown in figure 4. Magnitudes of the other models are similar on the same place, depending on the material properties and whether there is a

significant material transition in the neighbouring unit or not. For models having a lower Young's modulus, Shmin (Sh) and SH are lower, and the opposite for the models with the larger Young's modulus. Usage of variable material properties within the model units lead to different stress magnitudes within these units. The adjustment of the boundary conditions would always depend on which point of comparison would be chosen. Overall, the boundary conditions are defined by displacement, based on the reference model.

*I am afraid I do not fully understand the boundary conditions. What is the pre-stressed basic model, initial stress? Where is the virtual well? For which model does the boundary condition is calibrated? I imagine that different boundary conditions will be needed to fit the stress profile presented in Fig.4 depending on the rock properties. Is the boundary condition similar for all models? What is the impact of the boundary condition on the results? I agree that SHmax is more difficult to calibrate, however, it seems that the chosen value is significantly different than the one provided in the Brudy et al. (1997). In particular, there is no change in the stress regime in the case of Brudy et al. (1997). I think it is important to understand the impact of this change in the stress regime in the models and discuss it, as it is poorly constrained.*

KR: Before boundary conditions will be applied to the model, initial stress conditions are needed. These initial stress condition are a more complex issue. First of all, gravity acts instantaneous to a model, which would lead to significant settlement of the model. The initial stress condition avoids the settlement to a large extend. Furthermore, the observed ratio between SV (vertical stress)) and SH plus Sh are not a linear function with depth in contrast to simple isotropic or hydrostatic assumptions. The k-ratio ((SH+Sh)/2*SV) is much larger than one in the first hundred to thousand(s) of meters and becomes smaller than one in greater depth. This observation is supported by many data, but also by a semi-empirical function, provided by Sheorey(1994). His model is representative for an Earth without topography and tectonics. By settling the model without lateral shortening, but a large Poisson' ration under gravitational load, the derived stress state provides a quite good fit to Sheorey's profile as a function of depth and the Young's modulus. Therefore, the initial stress conditions provide a stress state with depth, which accounts broadly speaking for conditions without tectonic shortening. The models are pre-stressed in this way and underwent the tectonic component, or the difference between Sh and SH by the application of lateral boundary conditions. The subject of initial stress condition where kicked out, because reviewers usually are not interested in that subject and asks to skip that topic. Section 4.4 (line 223) is added to explains that subject.

The boundary conditions are simple, the model is shortened in north-south direction by 400 m and extended by 60 m in east-west direction, as shown in Fig 2. of the manuscript. The virtual well is in the centre of the model. The boundary conditions are calibrated on the reference model (Fig 4.). These boundary conditions are applied to all the different model runs. The usage of a different Young's modulus leads to a change of the magnitudes of SH and Sh. A larger Young's modulus will lead in larger stress magnitudes and vice versa. However, as mentioned earlier, a fit of a virtual well (stress profile) from a heterogeneous to a homogeneous model depends on the selected area or unit. Therefore, homogeneous boundary conditions are chosen. Less shortening/extension along the boundaries would lead to a lower difference between SH and Sh.

*The results concern the stress orientation at a depth of 1000 m below the surface, where there is a strike-slip fault regime according to the chosen boundary condition. How do the results change with depth and as a function of the stress regime? Why the results only concern the depth orientation at 1000 m? Also, do the rotations only occur in map view or is there also stress rotation in cross-section. In other terms, is the plan of observation presented comprise the principal stress?*

KR: The orientation of SH changes with depth, which is shown in Fig 12. The visualization of SH orientation is always in a depth of -1000m, to make it comparable. The scope of that study is to investigate the impact of the material properties and faults, as mentioned in the introduction. Due to the simplification of the model, I don't want to interpret the stress changes with depth too much in detail. However, the stress rotation decreases with depth (Fig. 12). Next to the material transition, SH and Sh becomes smaller near the surface and larger near the bottom of the model, within the stiffer units. In other words, the units with a lower Young's modulus causes a reduction of the stress magnitudes (SH and Sh) within the neighbouring stiffer units near the surface and increasing stress magnitudes in greater depth. It is not an issue of regime change. Magnitudes of SV, SH and Sh are pretty close to the principal stress magnitudes.

*The author tests separately different parameters: the density, the PR and the YM. The ranges of parameters tested seem correct. Models are designed to test each parameter individually. I think that this is a relevant method for a generic study. I am wondering however what is the geological meaning of this. In nature, these parameters can be interdependent. For example, a rock with a low density may have a low YM as well. Also, do the materials and discontinuities have constant properties with depth? How does this potentially impact the results of the models? I recommend the author to discuss further these points.*

KR: Sure, the tested material properties are not independent in nature. However, there is no linear relation between density, Young's modulus nor the Poisson's ratio. As shown by Fig. 3, density and Young's modulus for sediments are quite variable. For example, a limestone can have a smaller density, but larger Young's modulus as a shale. However, there is no geological meaning of the material variation, as it is a generic study. Sure, these properties will change with depth (or temperature) This is not included in the model. But usage of linearly increasing rock properties will affect the model results only slightly. Important is the effect of the material transition. For linear increasing density, Young's modulus or Poisson's ratio, the resulting effect remains similar. This subject is added in the discussion now.

*Section 5.*
*In the methodology, it is stated that the model is orientated relative to the North. Therefore, why no presenting the results with actual stress orientations rather than talking about clockwise and anti-clockwise rotation. Maybe this will help a little bit the reading.*

KR: I do not see the advantage of saying instead of clockwise or anticlockwise rotation to east or west.

*I found the results difficult to compare between the models. Maybe a cross-plot or a histogram comparing the various models could help.*

KR: A histogram is now included within the discussion section (Fig 13).

*I found that the results section lack of explanations of the behaviour of the model. For examples, what causes the rotation in the case of the density models? Why the models with the faults have little rotations. Is it because the faults are not critically stressed because they are not optimally orientated? More generally, I encourage the author to provide rationals for the observed behaviour.*

KR: Gravitational load is one of the major sources of stress in the Earth crust. This body force effects the magnitude of SV. However, due to the mechanical effect, which is specified by the Poisson's ratio, a locally increased SV lead to a local increase of SH and Sh, which can affect the stress pattern locally too.

The models with the faults have SH rotation next to the faults, as the principal stresses are always parallel or perpendicular to the fault (contact) like next to a free surface. However, observed stress orientation, which are presented in the figures are always placed away from the faults (>12.5 km), as I am only interested in the far field effects, not the near field effects. For sure, near field effects around faults would be a different subject, but not a subject of that study. The orientation is not optimal, but the chosen properties (C=0, FA~5°) are that low, that the units on each side are decoupled from each other for sure. In general, active faults do not have a large impact on the far field stress pattern at all, except close (~<10 km) to the discontinuity. That distance again depends on the Young's modulus. These subject are discussed in more detail in the discussion chapter.

*Section 6.*
*It seems that the author made a significant effort in reviewing previous studies on stress rotations. I will suggest providing a table summarizing this. This will help the reader and be an added value for the manuscript.*

KR: According to the suggestions, a table 1 with the previous studies is included.

*Concerning stress rotation and faults. Do we expect the stress rotations to be time dependant and change between when the fault is locked and when the fault slip? Concerning the results from the upper panel in Fig. 10. There is a significant difference between the behaviour in the shallow part of the model and the deeper part. Is this related to the boundary condition and the fact that the faults are critical stress in the strike-slip regime and locked in the normal faulting regime?*

KR: In the case of homogenous material properties, the far field stress pattern is independent whether a fault is locked or not, as shown by the model series with variable friction properties. The stress rotation is not time depended for the models, as elastic material properties are used. It's a function of the variable material properties and the boundary conditions. The faults in the models did not have any cohesion, the friction coefficient is quite low (0,1) which is a friction angle of about 5-6°. The different orientation of SH in the shallow and deeper parts are not affected by the boundary conditions.

The complex pattern in the upper cross-section (eEe) of Fig. 12 is a product of the fact, that no fault (or a locked fault) is included at the material transition. The pattern is a product of the interaction of the different Young's modulus in the neighbouring units. The soft units with a low Young's modulus (YM) will be deformed depending on the action boundary conditions. The stiff units with a higher YM cannot be deformed so easily. This transition leads to a compression of the stiff units next to the material transition, resulting in a rotation of SH.

This leads to a reduction of SH and Sh near the surface, next the material transition within the stiff units. This leads to the complex stress pattern. Active faults decouple that (|e|E|e|, Fig. 10). They balance the stress orientation (and the stress magnitudes to a certain point).

*I think that section 6.8 should be presented in the result section and not in the discussion. "The comparison will concern only the results of the models investigating the Young's modulus, as this material parameter have the strongest impact" but the model integrate densities and Poisson ratio according to table 2? Also, to help the reader, maybe the author could cross-plot the stress rotations from the model with the stress rotations from Fig. 1.*

KR: Yes, it is an option, to place section 6.8 in the result section. This has been done. Yes, the models apply all estimated material properties from the Variscan units in Germany (Table 2). It could be an option, to plot an average orientation from Fig 1 in comparison next to the result figure, allowing a better comparison. This has been implemented, see new figure 11.

*General organisation. Generally, the necessary pieces of information are provided in the manuscript, but I found that the paper lack clarity and organisation. The investigated geometry is interesting. I think however that the results depend on the chosen geometry, especially the strike and dip of the discontinuities/units and are therefore limited in scope. From my point of view, this work is both a generic study and a simplified study case. Accordingly, I recommend the author to re-organise their manuscript into a more classic scheme. For example (1) an introduction that merges the content of sections 1 and 2 after removing all unnecessary materials, with a review of stress rotation, a review of the key controls and a summary of the objectives (a generic study that discontinuity. aims to identify the key parameters and a case study in central Europe where there is good coverage of the stress field and a good knowledge of the regional geology to test the parameters). (2) Methodology. (3) Geological setting of the study case. (4) Result section divided into (i) generic models and (ii) a more realistic model. (4) Discussions centred on comparing the results with previous works.*

*Text. I think that the manuscript suffers from numerous English mistakes and unclear sentences. This sometimes obfuscates the message and I think that the manuscript, in general, will benefit from thoroughly polishing the text before publication. Several suggestions are provided below, but this is by no means an exhaustive list.*

*Technical comments:*
*L.31: "It was suggested, that" remove coma after suggested.*
*L.35: "sediments, and were" remove "and".*
*L:37-40: These sentences are not very clear. I am not sure to understand what "the assumed stress pattern (stress rotations)" refers to.*
*L.39: "can only partly explained" replace by " can only be partly explained".*
*L.43: "These 2-D models was" replace by "These 2-D models were".*
*L.43: "stress pattern" replace by "stress patterns".*
*L:43 ", applying" replace " by applying" and remove coma?*
*L.44-46: Maybe this could be simplified. For example: However, these 2D models cannot account for topography, crustal thickness and depth variability in stiffness and can overestimate horizontal stresses (Ghosh et al., 2006). Furthermore, none of these previous studies compared the impact of the influencing factors separately.*

KR: All proposed changes are implemented.

*L.47-48: Not clear. In this paper, we use a series of generic models to identify which properties can cause substantial stress rotations away from a material transition or a*

KR: This has been changed:

In this work, a series of large scale generic models is used to determine which properties can cause significant stress rotations at distance (>10 km) from material transitions or discontinuities. (line 49)

*L.49: "orientations" replace by orientation.*
*L.49: "north-south orientation" N-S orientation.*

KR: All proposed changes are implemented.

*L.51: Orientations are usually given with three digits. N030∘.*

KR: The three digit notation is useful for example sampling joint face data, to differentiate dip azimuth and dip (CLAR notation). In the most cases here, the relative rotation of the orientation is given. Therefore I prefer no changes on that notation.

*L.50-51: "The basement structures there are striking about 30° , which is perpendicular to the observed SHmax orientation" actually N030◦ is not perpendicular to N150◦, there is a 20◦ misfit.*

KR: This has been corrected.

The basement structures there are striking 45 to 60°, which is almost perpendicular to the observed SHmax orientation. (line 53)

*L.57: "The second major driver are" replace by "The second major driver is".*
*L.57-58: "Plate boundary forces where identified and derive deviatoric stresses" this sentence is not clear.*
*L.62 : "The most of these features" replace by "Most of these features".*
*L.62-135: see some corrections in the Specific comments section.*
*L.135: "both is not a subject of that study" replace by "both are not the subject of this study".*
*L.168: "between the horizontal stresses increases" remove s at increases.*

KR: Theses sentences are removed according previous suggestions.

*L.169: " is investigated" replace by "has been investigated"*
*L.191-192: "The explanation of that crustal structure are a cold, dense and slowly subsiding lithospheric root beneath the Alps (Müller and Zürich, 1984)." replace by "This crustal structure can be explained by: : :"*
*L. 193: "an subject" replace by "the subject"*

KR: All proposed changes are implemented.

*L.195: "which factors contributes" remove s at contributes.*

KR: This sentence is removed according previous suggestions.

*L.207: "topography has major effects" replace by "topography have major effects"*
*L.401-402: " The importance of stiffness differences is an result of other models too.." This sentence is not clear. "Similar impacts of stiffness contrast have been described in previous works: : :"*

KR: All proposed changes are implemented.

*L.415-417: These sentences are unclear.*

KR: This has been modified.

When comparing the stress rotation it is important to consider the respective depth (see Fig. 12). For example, data in the north-western Alps are dominant focal mechanisms and in the foreland of the central Alps, the majority of data are from wells, which are more shallow (Reinecker et al., 2010). (line 417)

*L.424: "this generic models" replace by "these generic models".*

KR: This has been modified.

*L.462: There is 6.8.1 but no 6.8.2?*

KR: The superfluous subheading was removed

*L.430: "indicates" replace by "indicate"*

KR: This has been modified.

*L.431: The work by Petit and Mattauer, (1995) concern mesoscale faults and I am not sure about this 2 km distance indicated here.*
*L.431: "is to" replace be "can be"*

KR: The content of that sentence is changed to clarify this.

Observations from meso-scale outcrops indicate stress perturbation within 2 km (Petit and Mattauer, 1995) or less than 1km to a fault (Rispoli, 1981); larger stress perturbation can be observe at the termination of the fault (2-3km). (line 453)

*L.466: "parameter have the" replace by "parameter has the"*
*L.477: "should deflected" replace by "should be deflected"*
*I hope this will help to improve the manuscript.*

KR: All proposed changes are implemented.

**Reviewer 2 (https://doi.org/10.5194/se-2020-129-RC2, 2020)**

*General comments*
*The article addresses an interesting problem of tectonic stress deviation due to the contrast in mechanical properties and fault motion. To explore this problem the Author designed models built of elastic and contact elements, loaded with body forces and tectonic strain. The models have a very simple structure and seem to be correctly constructed and solved. Some of the obtained results are interesting. However, this study, as well as the text itself, has many significant drawbacks, which I will focus on below. I do not mention language issues as I have no competence in this area.*

*Critical remarks:*
*Chapters 1. 2. and 3. Introduction*
*The Introduction is very long and exhausting, resembles an academic lecture. There are summarized various aspects of stress generation and measurement, very loosely related to the research subject. In my opinion, such chapters as 2.1, 2.2, 2.3 are not necessary and should be altogether shortened to one paragraph. On the other hand, the geodynamic context of stress rotation is poorly introduced. It would be better to present several natural examples of stress rotation and their possible reasons. The passages related to stress rotation in mechanical models can also be extended. Also chapters: 3.1. and 3.2 should be shortened significantly, as the European regional context, as it turns out at the end of the article, does not play an important role in the evaluation of the modeling results. Instead, more patients should be paid to the realistic crust profiles which determine ranges of material properties like density, stiffness, and to the effective friction coefficient of regional fault zones. I would suggest focusing on the key factors important for this modeling study.*

KR: Many of the sections are condensed now.

*Chapter 4. Model Setup*
*The structure of the model and boundary conditions are not sufficiently described. E.g. it is not clear how the faults terminate at the model boundary and what is a dip of them. In my opinion, the definition of constant elastic properties and density for the entire crustal thickness based on the property of rock present in the upper crust is a bad idea. Among the lithologies presented in Fig. 3 only granite can build the entire crust. Even the simple models should consider a realistic range of material properties, evaluated from typical lithologies of the Earth's crust. Neither in the introduction nor in the model setting chapter there is no reference to the realistic lithological profile of the Alpine foreland plate, based on geophysical constraints. Some references are in the discussion but without references to the lithological composition of the crust. The densities are given in Table 2 like 2.1. - 2.2 g/cm3 are typical for salt rock but for the crystalline crust !. The constant density across 30 km thick crust is also unrealistic. The results of modeling could be more significant when realistic and geophysically documented crust properties were used. The author should also justify such a dramatic change in YM across the entire crust. I can imagine that tenfold differentiation of effective stiffness can be produced by e.g. high heat flux variations, which are not present in the reference units. However, the assumption of elastic crust in the area with a moderate surface heat flow density is difficult to defend. The crust is probably rheologically layered, thus the larger part of the weaker crust unit is inelastic. Even accepting elastic simplification of the model the stiffness and structure of the model should follow from the realistic lithological and rheological profile of the crust.*

KR: Realistic models with a complex geometry, variable material properties, laterally and with depth, and a complex rheology are appropriate to reproduce observed mechanical features in detail. This is suitable for generating best-fit-models. However, the more 'adjusting screws' are technically implemented, the easier it becomes to achieve an optimal fit to the observation. To me the question

is: Are such best-fit-models suitable for identifying the most important parameters for stress rotation in the crust? Do such complex models give us a better understanding of the interaction of the properties used and discontinuities, which create or prevents stress rotation? For me, the answer is simple: No.

It has never been the aim of this study to present a model of the earth's crust that is as realistic as possible. It was mainly concerned with identifying the influence of density, elastic material properties (Young's modulus and Poisson's ratio) and discontinuities on the stress orientation, which deviates from the assumed stress orientation due to plate boundary forces. More important than the specific values are the differences in the properties themselves. Therefore, simple generic models are used, to test each parameter separately at first. Interaction of these parameters tested afterward.

*Chapter 5. Results of modeling*
*The results of modeling for variable PR, and density, even for unrealistically high contrast of parameters did not give significant results. Also, the stress rotations at faults are negligible, which is probably caused by their orientation under a high angle to Shmax which is not preferential for reactivation and by their ideal planar geometry.*

KR: May be these faults are not optimal oriented, but with the low friction coefficient (0.1) and zero cohesion, slip will happen anyway.

*The normal fault stress regime in the lower part of the model (mentioned by the author before) is additionally the reason why there is vanishing shear stress at the vertical plane of the fault.*

KR: For a vertical fault plane, the overall stress regime is not important, in contrast to the difference between the horizontal stresses (SH and Sh).

*However, the understanding of modeling results needs a better explanation of the fault implementation in the previous chapter. More significant but quite obvious results were obtained for YM variation, although the contrasts in this parameter are unrealistically high. The more significant modeling results require closer examination. It would be good to check the sensitivity of the model to changes in YM and present it on graphs or in the table. The most interesting results were obtained for combined Young modulus and fault slip. The results point that stress rotations between units of contrasting stiffness can be reduced by active faults. However in this case the analysis and presentation of fault displacement are necessary. When exploring this on realistic parameters and geometries the paper could be more interesting.*

KR: The YM variation is not unrealistic high to me, see Fig. 3. (Sorry for repetition: The focus of that study is on generic models.) The displacement along the faults depends on the model size for the modelling case. Important is the decoupling itself. Lateral offset is now mentioned in the result section.

*Chapter 6. Discussion*
*In chapter 6.1, the Author honestly states that the adopted assumptions regarding the material parameters do not match the model of the Earth's crust. The elastic thickness fits more closely with the lithosphere than the crust. So the Author is aware of the weaknesses of these models but why not translate it into a realistic model setup. In this chapter the result of stress changes with depth are presented, which better match chapter 5. Such analysis could be interesting, but unrealistic material properties and especially pure elastic mechanics make them not applicable to the Earth's crust. The interesting results of experiments with coupled faults and YM changes could be better*

*presented and the factors governing regularities in obtained results should be much deeper explored and explained.*

I am aware of the simplicity of the models used. Therefore, I have avoided to present too much details, which probably would pretend the use of realistic scenarios. Much more realistic models are needed to study such details.

*Chapter 7. Conclusions*
*In conclusion, the author admits that the modeling results do not reflect the stress rotation in the reference part of the Alpean foreland. Conclusions are very modest in comparison to the volume of the paper.*

KR: The major subject of that study are generic model. They are inspired my a specific region. The application of the simple model to the complex structures in the German Variscides is not very convincing. However, results of the generic models are very interesting to me. Therefore, the focus of the conclusion is on the interesting results.

*Some detailed remarks.*

*135 "Density contrast and topography" The topographic stress is a separate and wide subject, not investigated in this study. It is better to skip this issue instead of just put a number of references.*

KR: Topography and density contrast lead (partially) to the same effects for certain depths on the stress state.  Therefore, topographic effects are mentioned in connection with density contrasts.

*140 " stresses due to topography and crustal inhomogeneities are in the order of tens of MPa, 140 which are on a similar magnitude as the plate boundary forces". In this case, stresses should not be directly compared to forces.*

KR: This sentence has been clarified:

..., due to topography and crustal inhomogeneities are in the order of tens of MPa, which is in the order of stresses resulting from the plate boundary forces. (line 76)

*153 "Mechanical strength describes the material behaviour under the influence of stress and strain. The focus here is on elastic material properties, characterized by the Young's modulus and the Poisson's ratio." Please consider that elastic properties do not characterize strength.*

KR: The term 'strength' is not used any more in the context of the YM. It has been replaced by 'stiffness'

*168 "Small differences between the horizontal stresses increases the effect of faults on the local stress pattern" To some extent, because low differential stress means also low shear stress at the fault plane.*

KR: I agree for a strike slip regime. Small differences between horizontal stresses are not a sure indication for low differential stress in the case of a normal – or thrust faulting regime.

*368 "Fore sure it is really unlikely that such constant materials with such a thickness exists somewhere in the crust" YES for sure. Then why such unrealistic materials were tested?*

KR: As mentioned previously, the focus of that study is on generic models.

*371 "However, the overall geometry seems reasonable, as the brittle domain or elastic thickness of the crust (Te), which is a measure of the integrated strength of the lithosphere" Please consider that this is a lithosphere but not the crust itself. The upper mantle often contribute to this elasticity, then the mechanical properties assumed for the model are even less appropriate.*

KR: This is corrected now: 'elastic thickness of the lithosphere' instead of '… the crust'

*410 "The observed radial stress pattern southward of the Bohemian massif (Reinecker and Lenhardt, 1999) agrees 410 well with this study" As there is a lack of good examples of data constraining presented modeling results, this special case could be illustrated in the figure.*

KR: I think that the reference is sufficient.

*455 "The observed stress rotation strongly depends on the depth. In the soft units, SHmax rotates counter-clockwise near the surface (0 to -8 km). In contrast to that a clockwise rotation can be observed in greater depth (18−30 km)" Such an interesting result can be presented in more detail in Chapter 5. However, only the results from the upper part of the models are significant, due to the inelastic effects prevailing in the lower part.*

KR: I am aware of the simplicity of the models used. Therefore, I avoided to present too much details, which probably would pretend the use of realistic scenarios and e.g. realistic material variation with depth. Much more realistic models are needed to study such details. The mentioned observation is out of the primary scope of that study. So, it is impossible to follow up every new insight in detail.

*491 "but frankly speaking, the model results are not able to prove the significant influence of the material properties on the stress orientation for this region." That means, that the area selected for model evaluation is incorrect.*

KR: The generic models are inspired by a specific region, to investigate the interaction of variable material properties and the resulting stress orientation. When models are tested to specific regional settings, there is no correct or incorrect result.

*To summarize, the paper presents simple and unrealistic models of the Earth's crust. A large part of the text is too long or unnecessary, while the most interesting points are insufficiently described. I would recommend a major revision or rejection of this paper.*

---

## Author Response (AR2)

In the following, I will comment and reply to the review of Referee #1 from April the 22th, 2021.

To be able to distinguish well, I use the following colour code:
*Reviewer comments in italic and dark green (Ref#1)*
My comments in black and normal letters (KR)
Changes are indicated in dark blue

KR: First of all, I have to thank Referee #1 to read and review the manuscript again carefully and in detail. All proposed changes are well thought out. Therefore, the most suggestions have been adopted. However, two out of them cannot be implemented suitable. I will explain that in detail below.

*Ref#1: The author modifications improved the overall quality of the manuscript, the writing, and the structure. The author provided detailed replies to my previous comments. Furthermore, the modifications in Figure 11 and the new Figures (Table 1 and Figure 13) are valuable assets to this publication. However, in places, I found that incorporating further some of the contents of the author's replies within the manuscript could be valuable for the readers (see main comments below). Besides, although the text has been improved from the first version, I found that there are still some unclear sentences (see some potential suggestions in technical comments). For this reason, I suggest the publication of this manuscript after a Moderate revision.*

*Main comments*
*Ref#1: Table 1 is a summary of previous works from the literature. I think this table is a great asset in the publication. However, most of the data in this table show maximum values of rotations of 90, which is somehow not very informative. Maybe providing other values, like an average rotation or a rotation at a distance from the perturbation sources could be more informative. Besides, a value of 90 could also correspond to a stress permutation between Sh and SH, which can be related to changes in the magnitudes of the stress rather than the rotation of the stress. This may be somehow worth discussing to avoid confusion.*

KR: I understand quite well the intention of the referee. The most of the entries in the list uses numerical models, which are based on a certain mesh. The problem is, that the mesh resolution is poor, especially for the older publications. Next to other aspects, potential stress rotation depends on the amount of elements between a transition and the observation point. Therefore, providing a distance would be rather a measure for the mesh resolution than for the distance. The same applies to an average value or observed range or rotation. There is already an indication, whether the rotation is within or larger than 10 km away from the specific transition. Therefore, I propose to leave it as it is.
I agree, a stress rotation of 90° at a certain location is most probably the result of a magnitude change. But the spatial distributed gradual rotation of SH (e.g. Lund Snee&Zoback, 2016, 2020, Heidbach, 2018) could not be explained with that argument.

*Ref#1: The author clarifies the initial stress conditions by adding Section 4.4. The author also clarifies the boundary conditions in his reply and Section 6.2 of the manuscript. Even if this is now clearly stated in Section 6.2, I will suggest adding a few general sentences in section 4.5 for the readers. For example: "The boundary conditions are defined by displacement. They are calibrated on the reference model. The same boundary conditions are applied to all the different models."*

KR: The proposed change is now used.: '… which are technical applied by a defined lateral displacement. … a good fit of the reference model…. The determined boundary conditions are used for all models.

*Ref#1: The author answered the comments concerning the dip of the contacts between the units in his reply. Here again, maybe adding some of the information from the reply in Section 4.1 will be valuable for the readers. For example: "More complex geometries and variable dip angle may result in different stress patterns as the ones obtained for vertical discontinuities, however, studying such variability is beyond the scope of this paper."*

KR: The proposed change is now used.: "More complex geometries and variable dip angle may result in different stress patterns as the ones obtained for vertical discontinuities. However, studying such variability is beyond the scope of this study."

*The choice of the material properties is now more clearly discussed in Section 6.2 and the potential effect of failure are also discussed in Section 6.7.*
*I found the discussion interesting and detailed. But it is a bit difficult to follow, partly due to the numerous sections, which are sometimes very small. For simplification, maybe the author could group Sections 6.1 and 6.2. Similarly, maybe Section 6.4 could be incorporated in Section 6.5.*

KR: The proposed grouping of sections is now used.

*Technical comments*
*Ref#1: L.12: 'the horizontal stress orientation' maybe remove 'orientation'.*

KR: The proposed change is now used.

*Ref#1: L.13: 'in the order of up to 78' remove 'in the order of'.*

KR: The proposed change is now used.

*Ref#1: L.14: 'not only regions near the material transition (<10 km) are affected by this stress rotation' this is unclear.*

KR: "…even beyond the vicinity of the material transition (>10km)."

*Ref#1: L.232: See also Roche and Van der Baan 2015 and 2017 for references. (Roche, V., & Van der Baan, M. (2015). The role of lithological layering and pore pressure on fluid-induced microseismicity. Journal of Geophysical Research: Solid Earth, 120(2), 923-943.).*

KR: Both papers provide interesting work. However, to use the reference at that point would be incorrect. In the case, I understood correctly, Roche and van der Baan uses uniaxial and/or lithostatic stress conditions as initial stress state. The Poisson's ratio is varied vertically, depending on the lithology. The here used method varies the Poisson's ratio stepwise within the whole model.

*Ref#1: L.240: 'imagined' maybe 'virtual' is a better term.*

KR: The proposed change is now used.

*Ref#1: L.268: 'Mechanical' replace by 'Mechanically'.*

KR: The proposed change is now used.

*Ref#1: L.270: 'Therefore, the unit stiffness are from the deformable to the rigid ones: RHZ _ NPZ < STZ < MGCH _ MZ' This is not very clear. Also, replace 'stiffness by stiffnesses'.*

KR: This sentence has been re-written: 'Therefore, the unit stiffnesses are different: they are from slightly deformable to rigid in the following order RHZ ≈ NPZ < STZ < MGCH ≈ MZ.'

*Ref#1: L.278: 'no rotation is to observe' replace by 'no rotation is observed'.*

KR: The proposed change is now used.

*Ref#1: L.279: 'turns more counter-clockwise' remove 'more'.*

KR: The proposed change is now used.

*Ref#1: L.286: 'in the large density units' remove 'the'.*

KR: The proposed change is now used.

*Ref#1: L.300: 'Within the models having three' replace by 'For the models with three'.*

KR: The proposed change is now used.

*Ref#1: L.317: 'of a significant Young's modulus contrast with a cohesionless contact' replace by 'between a significant Young's modulus contrast and a cohesionless contact'.*

KR: The proposed change is now used.

*Ref#1: L. 321: 'point out significant larger rotation than for the stiff units' replace by 'shows larger rotations than the model with stiffer units'.*

KR: The proposed change is now used.

*Ref#1: L.324: 'In the models with the alternating stiffness with the low friction discontinuities (|E|e|E| and |e|E|e|) generate' replace by ' The models with alternating stiffnesses and low friction discontinuities (|E|e|E| and |e|E|e|) generate'.*

KR: The proposed change is now used.

*Ref#1: L.334: 'The same can be observed less pronounced in the observed SHmax orientation' replace by 'In Figure 11c, these areas show similar, but less pronounced, rotation of Shmax'.*

KR: The proposed change is now used.

*Ref#1: L.343: 'It is really unlikely that … as a result'. This is not clear, replace by 'Chosen properties are constant over a depth of 30 km, which is unlikely. Even for a given lithology, the properties can change with depth, as a result of'.*

KR: The proposed change is now used.

*Ref#1: L.394: 'only 31 rotation' 'only a 31 rotation'.*

KR: The proposed change is now used.

*Ref#1: L.404: remove capital at 'model'.*

KR: The proposed change is now used.

*Ref#1: Figure 3: 'Poissons ratio' replace by Poisson's ratio*

KR: The proposed change is now used.

*Ref#1: Figure 12: 'counterbalances' replace by 'counterbalance'*

KR: The proposed change is now used.

*Ref#1: Figure 13: replace max. rotation [°].*

KR: The proposed change is now used.